Manuscript prepared for Atmos. Chem. Phys.
with version 2014/09/16 7.15 Copernicus papers of the LaTeX class copernicus.cls.
Date: 15 December 2016

# Directional, Horizontal Inhomogeneities of Cloud Optical Thickness Fields Retrieved from Ground-Based and Airborne Spectral Imaging

Michael Schäfer[1], Eike Bierwirth[1,2], André Ehrlich[1], Evelyn Jäkel[1],
Frank Werner[3], and Manfred Wendisch[1]

[1]Leipzig Institute for Meteorology, University of Leipzig, Leipzig, Germany
[2]now at: PIER-ELECTRONIC GmbH, Nassaustr. 33–35, 65719 Hofheim-Wallau, Germany
[3]Joint Center for Earth Systems Technology, University of Maryland, 5523 Research Park Drive
320, Baltimore, MD 21228

*Correspondence to:* Michael Schäfer (michael.schaefer@uni-leipzig.de)

**Abstract.** Clouds exhibit distinct horizontal inhomogeneities of their optical and microphysical properties, which complicate their realistic representation in weather and climate models. In order to investigate the horizontal structure of cloud inhomogeneities, two-dimensional (2D) horizontal fields of optical thickness ($\tau$) of subtropical cirrus and Arctic stratus are investigated with a spatial resolution of less than 10 m. The 2D $\tau$-fields are derived from (a) downward (transmitted) solar spectral radiance measurements from the ground beneath four subtropical cirrus clouds, and (b) upward (reflected) radiances measured from aircraft above ten Arctic stratus clouds. The data were collected during two field campaigns: (a) Clouds, Aerosol, Radiation, and tuRbulence in the trade wInd regime over BArbados (CARRIBA), and (b) VERtical Distribution of Ice in Arctic clouds (VERDI). One-dimensional (1D) and 2D autocorrelation functions, as well as power spectral densities are derived from the retrieved $\tau$-fields. The typical spatial scale of cloud inhomogeneities are quantified for each cloud case. Similarly, the scales at which three-dimensional (3D) radiative effects influence the radiance field are identified. In most of the investigated cloud cases considerable cloud inhomogeneities with a prevailing directional structure are found. In these cases, the cloud inhomogeneities favour a specific horizontal direction while across this direction the cloud is of homogeneous character. The investigations reveal that it is not sufficient to quantify horizontal cloud inhomogeneities by 1D inhomogeneity parameters; 2D parameters are necessarily required.

## 1 Introduction

The globally and annually averaged cloud cover is in the range of about $70\,\%$ (Rossow and Schiffer, 1999). Because of this high percentage, and their important effects on the Earth's radiation budget, clouds need to be considered as an important regulator of the Earth's climate (Albrecht, 1989; Loeb et al., 2009). Clouds scatter and absorb solar radiation in the wavelength range from $0.2\,\mu m$ to $5\,\mu m$; they emit and absorb terrestrial radiation from $5\,\mu m$ to $50\,\mu m$. Although clouds have been studied for several decades, they are still poorly represented in weather and climate models (Shonk et al., 2011). The latest report of the Intergovernmental Panel on Climate Change (IPCC, 2013) classifies cloud effects as one of the largest uncertainties in climate simulations, significantly contributing to problems in the determination of the Earth's energy budget (Stocker et al., 2013). These issues partly arise from an unrealistic representation of complex horizontal cloud structures and from cloud-radiation feedback processes that control the cloud evolution (Stephens, 2005; Shonk et al., 2011). Therefore, the representation of cloud inhomogeneities needs to be become more realistic (Shonk et al., 2011). This is particularly important, because changes of cloud properties may have serious consequences on the interaction of clouds with radiation (Slingo, 1990).

Several independent studies investigated the influence of the plane-parallel assumption on cloud retrievals (e.g. Cahalan, 1994; Loeb and Davies, 1996; Marshak et al., 1998; Zinner et al., 2006; Varnai and Marshak, 2007). They found that the magnitudes of model biases are related to the degree of horizontal photon transport. In 1D radiative transfer simulations clouds are divided into separate vertical columns with horizontal homogeneous optical and microphysical properties (independent pixel approximation, IPA). However, horizontal photon transport cannot be neglected in case of inhomogeneous clouds. Additionally, multiple scattering due to 3D microphysical cloud structures smooth the horizontal radiation field. On small scales, this limits the accuracy of IPA. For example, Cahalan (1994) and Marshak et al. (1995) revealed discrepancies for individual pixel radiances exceeding $50\,\%$ due to a plan-parallel bias.

High ice clouds (cirrus) and Arctic stratus exhibit horizontal inhomogeneities at different horizontal scales. Both cloud types can either warm or cool the Earth's climate system, depending on their optical and microphysical properties and the meteorological conditions. For example, Choi and Ho (2006) reported for tropical regions a positive (warming) net radiative effect of cirrus for a cirrus optical thickness ($\tau_{ci}$) of less than 10, but a cooling effect for $\tau_{ci} > 10$. For Arctic stratus, Wendisch et al. (2013) showed that for low surface albedo ($\alpha_s$) and low solar zenith angle ($\theta_0$), the cloud cools the sub-cloud layer. With increasing $\alpha_s$ and increasing $\theta_0$, the cooling effect of the low–level cloud turns into a warming. Both, clouds and surface reflection properties can vary significantly and in different horizontal scales.

In many remote-sensing applications clouds are assumed as plane-parallel (Francis et al., 1998; Iwabuchi and Hayasaka, 2002; Garrett et al., 2003), which may introduce biases into the modeled radiation budget (Shonk et al., 2011). For example, in the cases of cirrus, Carlin et al. (2002) found

a plane-parallel cirrus albedo bias of up to $25\,\%$ due to spatial cirrus inhomogeneity. For Arctic stratus over variable sea-ice surfaces, Rozwadowska and Cahalan (2002) reported a plane-parallel albedo bias of less than $2\,\%$, but an absolute value of the transmittance bias that can exceed $10\,\%$.

3D Monte Carlo radiative transfer simulations account for horizontal photon transport (Barlakas et al., 2016). However, they are costly in terms of computation time and memory (Huang and Liu,
2014). This renders Monte Carlo radiative transfer simulations inappropriate for the application in operational or global models. Other approaches introduce Monte Carlo integration of independent column approximation (McICA), as proposed by Pincus et al. (2003). McICA is a computational efficient technique for computing domain-averaged broadband radiative flux densities in vertically and horizontally variable cloud fields (Pincus et al., 2003). Improvements compared to the plane-
parallel assumption are achieved with this approach, but results are still not as accurate as those from 3D Monte Carlo models. To reduce uncertainties associated with the 1D plane-parallel assumption, Huang and Liu (2014) apply spatial autocorrelation functions of cloud extinction coefficients to capture the net effects of sub-grid cloud interactions with radiation. With several orders less computation time, this approach reproduces 3D Monte Carlo radiative transfer simulations with an
accuracy within $1\,\%$. However, Huang and Liu (2014) assumed perfect knowledge about the spatial correlation functions of cloud extinction coefficients, which underlines the need for measurements of comparable resolved inhomogeneity measures.

General circulation or numerical weather forecast models require sub-grid scale parameterizations of, e.g., cloud structures, liquid water content (LWC), and/or ice water content (IWC) (Huang and
Liu, 2014). In reality, cloud structures reveal spatial features down to distances below the meter scale (Pinsky and Khain, 2003). Therefore, measurements with appropriate spatial and temporal resolution have to be conducted in order to derive the needed parameterizations. The required measurements include cloud altitude (temperature), its geometry (vertical and horizontal extent), and cloud microphysical properties (e.g., LWC, IWC, droplet size, ice crystal size and shape distributions).

Cloud inhomogeneities often exhibit a typical directional structure (e.g., induced by the prevailing wind). In such a case, 1D observations with LIDAR (light detecting and ranging) or point spectrometers can lead to an underestimation or overestimation of the degree of cloud inhomogeneity of the whole cloud scene. For example, a cloud with a rather inhomogeneous character may be classified as horizontally homogeneous (underestimation of inhomogeneity), if the dominating cloud structure
has the same orientation as the cloud observational path. Contrary, the cloud inhomogeneity would be overestimated if the cloud is scanned perpendicular to the major directional structure. Therefore, 2D observations are a useful tool to avoid such misinterpretations of cloud inhomogeneity.

In this paper, horizontal $\tau$ fields retrieved from solar spectral radiance measurements are analyzed to quantify horizontal inhomogeneities of two cloud types; subtropical cirrus and Arctic stratus. The
information content of 1D and 2D approaches on cloud inhomogeneity analysis are compared to identify their scientific value and limits. In Sect. 3, a statistical evaluation of the horizontal inho-

mogeneity of the fields of $\tau$ is presented using common 1D inhomogeneity parameters from the literature. Those bulk properties are valid to quantify the overall cloud inhomogeneity, but cannot reproduce spatial inhomogeneities of the cloud field. In Sect. 4, the derived bulk properties from the 1D inhomogeneity parameters are compared to 1D and 2D autocorrelation functions. Finally, in Sect. 5, 1D and 2D Fourier analysis is used to investigate the effect of horizontally cloud inhomogeneities on radiative transfer.

## 2   Data Set: 2D-Fields of Cloud Optical Thickness

Data from two international field campaigns have been analyzed: the Clouds, Aerosol, Radiation, and tuRbulence in the trade wInd regime over BArbados (CARRIBA, Siebert et al., 2013; Schäfer et al., 2013) campaign performed on Barbados in April 2011, and the VERtical Distribution of Ice in Arctic clouds (VERDI, Schäfer et al., 2015) observations carried out in Inuvik, Canada in May 2012. Two-dimensional (2D) fields of downward and upward solar spectral radiances ($I_\lambda^\downarrow$, $I_\lambda^\uparrow$) were measured from the ground (CARRIBA) and from an aircraft (VERDI). The imaging spectrometer AisaEAGLE (manufactured by Specim Ltd., Finland, Hanus et al., 2008; Schäfer et al., 2013, 2015) was used for the measurements. It is a single-line sensor with a field of view (FOV) of 37° and 1024 spatial pixels detecting radiation in the wavelength range from 400 nm to 970 nm with a spectral resolution of 1.25 nm full width at half maximum (FWHM). The 2D scans of the cloud scenes are generated from sequential (4 Hz to 30 Hz frame rate) measurements of the single sensor-line, while the target (cloud) moves with the wind (ground-based) or the flying aircraft across this sensor line. Adding up all measured lines behind each other, the 2D scan evolves as an image with a spatial (number of sensor pixels) and temporal (number of recorded frames) axis. Applying the known geometry, integration time, cloud and aircraft velocities, the axes dimensions can be transferred into distances. The 2D images evolved either from the heading of the clouds above the sensor line (ground-based), or by the movement of the sensor-line itself above the clouds (airborne). The imaging spectrometer was characterized and calibrated in the laboratory to transform the AisaEAGLE's raw data (12-bit digital numbers) into radiance. The procedure of data evaluation (calibrations, corrections) follows the methods described by Bierwirth et al. (2013) and Schäfer et al. (2013, 2015).

As proposed by Marshak et al. (1995), Oreopoulos et al. (2000), or Schröder (2004), horizontal cloud inhomogeneities are studied by scale analysis of cloud-top reflectances. However, radiance measurements include the information of the scattering phase function (e.g., forward/backward scattering peak, halo features) in the measured fields of radiance (Schäfer et al., 2013). To avoid artefacts in the scale analysis resulting from such features, parameters that are independent on the directional scattering of the cloud particles have to be analyzed. The cloud optical thickness $\tau$ does not include the fingerprint of the scattering phase function. Therefore, the ground-based and airborne measured fields of $I_\lambda^\downarrow$ (CARRIBA) and $I_\lambda^\uparrow$ (VERDI) were used to retrieve horizontal fields of $\tau$ with a spatial

resolution of less than 10 m. The retrieved fields of $\tau$ were then applied to investigate horizontal cloud inhomogeneities of subtropical cirrus (index ci) and Arctic stratus (index st).

Simulations are performed with the radiative transfer solver DISORT 2 (Discrete Ordinate Radiative Transfer). Input parameters such as cloud optical properties, aerosol content and spectral surface albedo are provided by the library for radiative transfer calculations (libRadtran, Mayer and Kylling, 2005). The required profiles of thermodynamic parameters are derived from measurements from radiosondes and/or dropsondes. Despite of assuming plane-parallel clouds in the simulations, the investigation of 3D radiative effects is still possible using the retrieved fields of $\tau$, but directional features related to the scattering phase function are avoided. $I_\lambda^\downarrow$ and $I_\lambda^\uparrow$ were simulated as a function of values of $\tau_{ci}$ and $\tau_{st}$, respectively. The simulations were performed for all scattering angles within the FOV of AisaEAGLE. Thus, simulated grids of possible $I_\lambda^\downarrow$ and $I_\lambda^\uparrow$ and corresponding $\tau_{ci}$ and $\tau_{st}$ are available for each time step of the measurements and each spatial pixel. The retrieved $\tau_{ci}$ and $\tau_{st}$ are derived by interpolating the simulated $I_\lambda^\downarrow$ and $I_\lambda^\uparrow$ to the measured value for each spatial pixel using a linear interpolation. More detailed descriptions and sensitivity tests of the applied retrieval procedures are reported by Schäfer et al. (2013) for subtropical cirrus and by Bierwirth et al. (2013) as well as Schäfer et al. (2015) for Arctic stratus. Fields of cloud optical thickness are derived for four subtropical cirrus cases ($\tau_{ci}$) and ten Arctic stratus cases ($\tau_{st}$). Subsequently, those fields of $\tau_{ci}$ and $\tau_{st}$ are used to investigate and quantify horizontal cloud inhomogeneities.

Table 1 summarizes the statistical parameters of the four retrieved fields of $\tau_{ci}$ (Ci-01 to Ci-04) and the ten retrieved fields of $\tau_{st}$ (St-01 to St-10). Figure 1 illustrates example cutouts for cases St-04 and St-07, both characterized by a measurement duration of 60 s. Table 1 further provides information on the measurement time, cloud altitude ($h_{cld}$), field size (swath, length), and average as well as standard deviation of $\tau_{ci}$ ($\bar{\tau}_{ci} \pm \sigma_{\tau,ci}$) and $\tau_{st}$ ($\bar{\tau}_{st} \pm \sigma_{\tau,st}$). The sampled subtropical cirrus fields of about 13-44 km length and 7-8 km width are determined by the time of observation and the swath covered by AisaEAGLE. For the Arctic stratus cases the average swath of the covered cloud fields has a size close to 1.3 km. The length varies from 4 km to up to 26 km. Thus, for CARRIBA and VERDI sufficiently large areas of the clouds are covered to provide a statistically firm analysis of $\tau_{ci}$ and $\tau_{st}$ and to investigate their horizontal inhomogeneities.

## 3   1D Inhomogeneity Parameters

The standard deviation $\sigma_\tau$ of the cloud optical thickness does not allow a comparison between cases with different average cloud optical thickness $\bar{\tau}$. A cloud with higher $\bar{\tau}$ can exhibit a higher standard deviation. Therefore, similarly to the studies by Barker et al. (1996), who used ratios between mean $\tau$ and the variance of $\tau$, Davis et al. (1999a) and Szczap et al. (2000) utilized the normalized inhomogeneity measure $\rho_\tau$ to quantify the horizontal inhomogeneity of $\tau$. It is defined by the ratio of $\sigma_\tau$

**Table 1.** Label, measurement period (day and time of day in UTC), cloud top altitude, pixel size (width, length), domain size (swath, length), and average and standard deviation ($\bar{\tau} \pm \sigma_\tau$) of the retrieved fields of $\tau_{ci}$ and $\tau_{st}$ from ground-based measured CARRIBA (Ci-01 to Ci-04) and airborne measured VERDI (St-01 to St-10) cases. The flight altitude for each VERDI case is at 2920 m. The right three columns include calculated 1D inhomogeneity parameters ($\rho_\tau$, $S_\tau$, $\chi_\tau$) of the retrieved fields of $\tau$. They are discussed in Sect. 3.

| Case | Day | Time (UTC) | $h_{\mathrm{cld}}$ (km) | Pixel Size (m) | Domain (km) | $\bar{\tau} \pm \sigma_\tau$ | $\rho_\tau$ | $S_\tau$ | $\chi_\tau$ |
|------|-----|-----------|------------------|----------------|-------------|-----------------------|--------|--------|--------|
| Ci-01 | 9 April 2011 | 13:26:26 – 13:37:13 | 11-15 | ≈ 7.1 x 4.8 | 7.3 x 15.6 | 0.41 ± 0.17 | 0.40 | 0.19 | 0.92 |
| Ci-02 | 16 April 2011 | 13:44:29 – 14:17:42 | 12-15 | ≈ 7.8 x 5.1 | 8.0 x 40.5 | 0.28 ± 0.09 | 0.35 | 0.15 | 0.94 |
| Ci-03 | 18 April 2011 | 13:43:56 – 14:17:13 | 13-15 | ≈ 8.4 x 5.5 | 8.6 x 44.1 | 0.20 ± 0.03 | 0.17 | 0.08 | 0.99 |
| Ci-04 | 23 April 2011 | 16:45:10 – 17:03:12 | 11-14 | ≈ 7.1 x 3.1 | 7.3 x 13.3 | 0.05 ± 0.04 | 0.91 | 0.48 | 0.63 |
| St-01 | 14 May 2012 | 20:31:47 – 20:36:31 | ≤ 0.90 | ≈ 2.7 x 2.5 | 1.34 x 21.28 | 9.93 ± 1.89 | 0.19 | 0.08 | 0.98 |
| St-02 | 14 May 2012 | 20:38:04 – 20:42:09 | ≤ 0.85 | ≈ 2.8 x 2.8 | 1.37 x 20.83 | 7.82 ± 2.01 | 0.26 | 0.11 | 0.97 |
| St-03 | 14 May 2012 | 20:53:26 – 20:58:30 | ≤ 0.85 | ≈ 2.8 x 3.0 | 1.37 x 26.85 | 3.82 ± 1.33 | 0.34 | 0.20 | 0.92 |
| St-04 | 15 May 2012 | 18:41:53 – 18:43:58 | ≤ 1.00 | ≈ 2.6 x 2.8 | 1.27 x 10.50 | 14.34 ± 2.54 | 0.18 | 0.08 | 0.98 |
| St-05 | 15 May 2012 | 21:05:10 – 21:09:24 | ≤ 0.93 | ≈ 2.7 x 2.7 | 1.32 x 20.22 | 6.35 ± 0.97 | 0.15 | 0.07 | 0.99 |
| St-06 | 16 May 2012 | 19:10:56 – 19:15:56 | ≤ 1.00 | ≈ 2.6 x 2.6 | 1.27 x 23.10 | 6.52 ± 1.48 | 0.23 | 0.11 | 0.97 |
| St-07 | 17 May 2012 | 16:53:23 – 16:56:06 | ≤ 0.25 | ≈ 3.6 x 2.6 | 1.75 x 12.74 | 3.04 ± 0.66 | 0.22 | 0.11 | 0.97 |
| St-08 | 17 May 2012 | 17:00:59 – 17:06:15 | ≤ 1.00 | ≈ 2.6 x 2.7 | 1.27 x 25.65 | 5.48 ± 1.84 | 0.34 | 0.15 | 0.95 |
| St-09 | 17 May 2012 | 17:09:28 – 17:10:38 | ≤ 2.25 | ≈ 1.0 x 2.6 | 0.48 x 5.46 | 7.07 ± 1.41 | 0.20 | 0.09 | 0.98 |
| St-10 | 17 May 2012 | 18:49:26 – 18:50:16 | ≤ 0.23 | ≈ 3.6 x 2.7 | 1.76 x 4.10 | 4.15 ± 0.67 | 0.16 | 0.08 | 0.99 |

and the average value $\bar{\tau}$ of the corresponding sample:

$$\rho_\tau = \frac{\sigma_\tau}{\bar{\tau}}. \tag{1}$$

A homogeneous cloud is characterized by $\rho_\tau = 0$. Increasing values of $\rho_\tau$ indicate rising cloud inhomogeneity. However, $\rho_\tau$ has no predefined upper limit, which might lead to misinterpretations
in a variability analysis. This renders $\rho_\tau$ not as a quantitative, but qualitative measure only. Therefore, Davis et al. (1999a) and Szczap et al. (2000) convert the relative variability $\rho_\tau$ into the inhomogeneity parameter $S_\tau$ as follows:

$$S_\tau = \frac{\sqrt{\ln(\rho_\tau^2 + 1)}}{\ln 10}. \tag{2}$$

In case of a log-normal frequency distribution of $\tau$, $S_\tau$ is proportional to to $\rho_\tau$. This is because
the reflected/transmitted radiance is approximately linear to $\log \tau$ for moderate $\tau$ (for $\log \tau = 0.5$-$1.5$ with $\tau \approx 3$-$30$). Without net horizontal photon transport, moments of reflected/transmitted radiance are closely linked with moments of $\log \tau$ rather than moments of $\tau$ (Iwabuchi and Hayasaka, 2002). Therefore, $S_\tau$ quantifies the degree of cloud inhomogeneity.

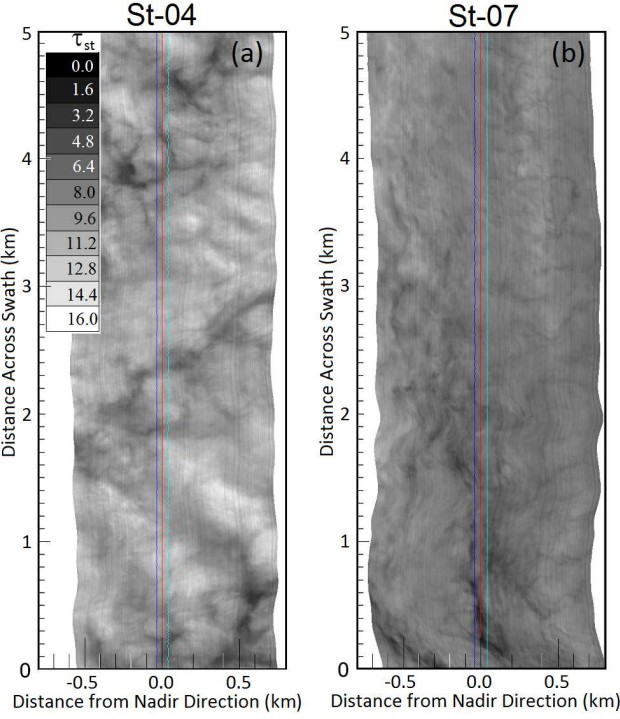

**Figure 1. (a)** Georeferenced field of $\tau_{\mathrm{st}}$. Cutout from measurement case St-04 with 60 s measurement duration. The dark–blue, red, and light–blue lines illustrate the nadir viewing direction in a range of $\pm\,1°$. **(b)** Same as (a) for case St-07.

Oreopoulos and Cahalan (2005) investigated the inhomogeneity parameter $\chi_\tau$, first introduced by
175 Cahalan (1994). $\chi_\tau$ is defined as the ratio of the logarithmic and linear average of a distribution of $\bar\tau$:

$$\chi_\tau = \frac{\exp(\overline{\ln\tau})}{\bar\tau}, \tag{3}$$

The 1D inhomogeneity parameter $\chi_\tau$ ranges between 0 and 1, with values close to unity indicating horizontal homogeneity, and values approaching zero characterizing high horizontal inhomogeneity.
Oreopoulos and Cahalan (2005) state that the reflected solar flux is approximately a linear function of the logarithm of $\tau$ for a wide range of $\tau$ ($\approx 3$ to $\approx 30$, depending on $\theta_0$). Thus, the logarithmically averaged $\tau$ provides a way to account for cloud inhomogeneity effects in plane-parallel radiative transfer calculations using $\chi_\tau$ as a scaling factor with which $\tau$ needs to be multiplied to approximate the IPA albedo.
The three 1D inhomogeneity parameters $\rho_\tau$, $S_\tau$, and $\chi_\tau$ are calculated for each retrieved field of $\tau_{\mathrm{ci}}$ and $\tau_{\mathrm{st}}$ from the CARRIBA and VERDI campaigns. The results are listed in the right three columns of Tab. 1. When comparing them to literature values one has to keep in mind that cloud inhomogeneities appear on different spatial scales. E.g., cloud fields may change on synoptic scales

($\approx 100$ km) or dynamic scales (10–100 m) depending on the cloud type. Therefore, inhomogeneity parameters depend on the pixel and domain size of the analyzed cloud fields. The larger the domain size or the smaller the pixel size is, the broader the probably density function of the cloud parameter may become. Therefore, a comparison of different cloud cases is only valid when pixel size and cloud domain are in the same range.

The subtropical cirrus cases observed during CARRIBA show $\rho_\tau$ in the range of 0.17–0.91, while $S_\tau$ is in the range of 0.08–0.48. The largest values of $\rho_\tau$ and $S_\tau$ are found for Ci-04, the lowest for Ci-03. The values for $\rho_\tau$ and $S_\tau$ show that the subtropical cirrus of Ci-02 and Ci-03 was quite homogeneous, whereas that of Ci-01 and Ci-04 was rather inhomogeneous. For the ten Arctic stratus cases, $\rho_\tau$ and $S_\tau$ are in the range of 0.15–0.34 and 0.07–0.20, respectively. For stratocumulus (6.9 km domain with 15 m horizontal resolution) Zuidema and Evans (1998) quantified the inhomogeneity of $\tau$ with $S_\tau = 0.1$–0.3. Iwabuchi (2000) and Iwabuchi and Hayasaka (2002) investigated the inhomogeneity of $\tau$ for overcast boundary layer clouds from a visible-wavelength moderate-resolution (about 1 km) sensor and found values of $S_\tau = 0.03$–0.3, which leads to $\rho_\tau = 0.07$–0.78. Thus, considering the different pixel and domain sizes, the derived values from CARRIBA and VERDI compare well with those reported by Zuidema and Evans (1998), Iwabuchi (2000), and Iwabuchi and Hayasaka (2002). Among all ten cases, $\rho_\tau$ and $S_\tau$ indicate case St-03 and St-08 to be more inhomogeneous.

For CARRIBA, the values of $\chi_\tau$ range from 0.63 to 0.99, indicating a rather inhomogeneous cirrus for Ci-04 and quite homogeneous cirrus during the other days. In contrast to the results for $\rho_\tau$, $S_\tau$, and $\chi_\tau$ indicate that the subtropical cirrus of Ci-01 is less inhomogeneous. The calculated values of $\chi_\tau$ for the retrieved fields of $\tau_{st}$ from the VERDI campaign yield values larger than 0.9 in each case, with lowest values for case St-03 and St-08, which were already indicated by $\rho_\tau$ and $S_\tau$ to be more inhomogeneous. Using the Moderate Resolution Imaging Spectroradiometer (MODIS), depending on cloud type, cloud phase, surface type, season, and time of day, Oreopoulos and Cahalan (2005) estimate the range of $\chi_\tau$ from $\approx 0.65$ to 0.8 at spatial scales of $1° \times 1°$.

The 1D inhomogeneity parameters $\rho_\tau$, $S_\tau$, and $\chi_\tau$ are easy to calculate and suitable for being implemented in simulations that assume horizontally homogeneous clouds to achieve more realistic results. They do not provide a measure of the directional variability of the inhomogeneities. However, different clouds exhibit preferred horizontal inhomogeneity patterns and typical features. For example, the clouds observed during CARRIBA and VERDI are different in terms of cloud altitude, structure, phase, particle size and shape, although $\rho_\tau$, $S_\tau$, and $\chi_\tau$ yield comparable values (compare Fig. 1 and Tab. 1). Therefore, not only the horizontal inhomogeneity, but also the spatial coherence of cloud inhomogeneity parameters and their directional dependence need to be investigated (Hill et al., 2012).

## 4 Spatial 1D and 2D Autocorrelation Functions and Decorrelation Length

The 2D autocorrelation function $P_\tau(L_x, L_y)$ is calculated in two horizontal dimensions at fixed distances (pixel-lags) $L_x$ and $L_y$, which are derived as integer multiples of the equidistant sample intervals $x_j$ and $y_k$ of the 2D fields of $\tau$ (Marshak et al., 1998). The maximum pixel-lags $L_x$ and $L_y$ are given by the number of pixels $n$ and $m$ of the 2D fields. Here, with $n$ and $m$ equidistant measurement intervals $x_j$ and $y_k$, $P_\tau(L_x, L_y)$ for 2D fields of $\tau$ is calculated by:

$$230 \quad P_\tau(L_x, L_y) = \frac{\sum\limits_{j,k+1}^{n,m} [\tau(x_j + L_x, y_k + L_y) - \bar\tau] \cdot [\tau(x_j, y_k) - \bar\tau]}{\sum\limits_{j,k+1}^{n,m} [\tau(x_j, y_k) - \bar\tau]^2}. \quad (4)$$

Here, $\tau(x_j, y_k)$ is the cloud optical thickness observed at the reference position, and $\tau(x_j + L_x, y_k + L_y)$ is the cloud optical thickness at pixel-lag $L_x$ and $L_y$. The autocorrelation function $P_\tau(L_x, L_y)$ yields values between -1 and 1, 1 representing a perfect positive correlation (e.g., for a spatial shift equal to zero); a value of -1 is a perfect negative correlation and 0 indicates no correlation. Thus, spatial autocorrelation functions quantify the degree of similarity between spatially distributed neighbouring samples. Usually, $\tau$ values in close horizontal distance reveal similar values, while cloud pixels at larger distances may show significantly different values of $\tau$, depending on the cloud heterogeneity. Here, only the degree of correlation matters; the positive or negative sign of the autocorrelation result is of less importance. To avoid misinterpretations with the sign, the squared autocorrelation function $P_\tau^2(L_x, L_y)$ is used here.

Figures 2b and 2d show examples of $P_\tau^2(L_x, L_y)$ in a 2D plot for $L_x = $ -250 to $L_x = 250$ and $L_y = $ -250 to $L_y = 250$, calculated for a selected area (500 by 500 Pixels, Figs.2a and 2b) of the cirrus fields from case Ci-01 and Ci-03 with $L_x = 250$ and $L_y = 250$. The positive and negative signs of $L_x$ and $L_y$ in Fig. 2b and 2d illustrate the direction into which the particular image is shifted against itself to derive the depicted autocorrelation coefficients. Both cases show a different pattern of $P_\tau^2(L_x, L_y)$ with increasing absolute value of $L_x$ and $L_y$. While Ci-01 shows a circular spot indicating a symmetry independent on direction, Ci-03 displays high correlation factors for all considered $L_y$ within a range of $L_x$=-50 to $L_x$=50. This pattern indicates a homogeneous cloud structure along the y axis while the $\tau$ field along the x-axis is heterogeneous. The magnitude of decrease of $P_\tau^2(L_x, L_y)$ with increasing $L_x$ and $L_y$ depends on the horizontal structure of the cloud inhomogeneities. The $P_\tau^2(L_x, L_y)$ calculated from Ci-01 (Fig. 2b) show a decrease, independent of the direction. In contrast, the $P_\tau^2(L_x, L_y)$ calculated from Ci-03 (Fig. 2d) show a directional dependence.

The squared spatial autocorrelation functions $P_\tau^2(L_x, L_y)$ are used to calculate the decorrelation length $\xi_\tau = \sqrt{L_x^2 + L_y^2}$ implicitly defined by:

$$P_\tau^2(\xi_\tau) = 1/e^2. \quad (5)$$

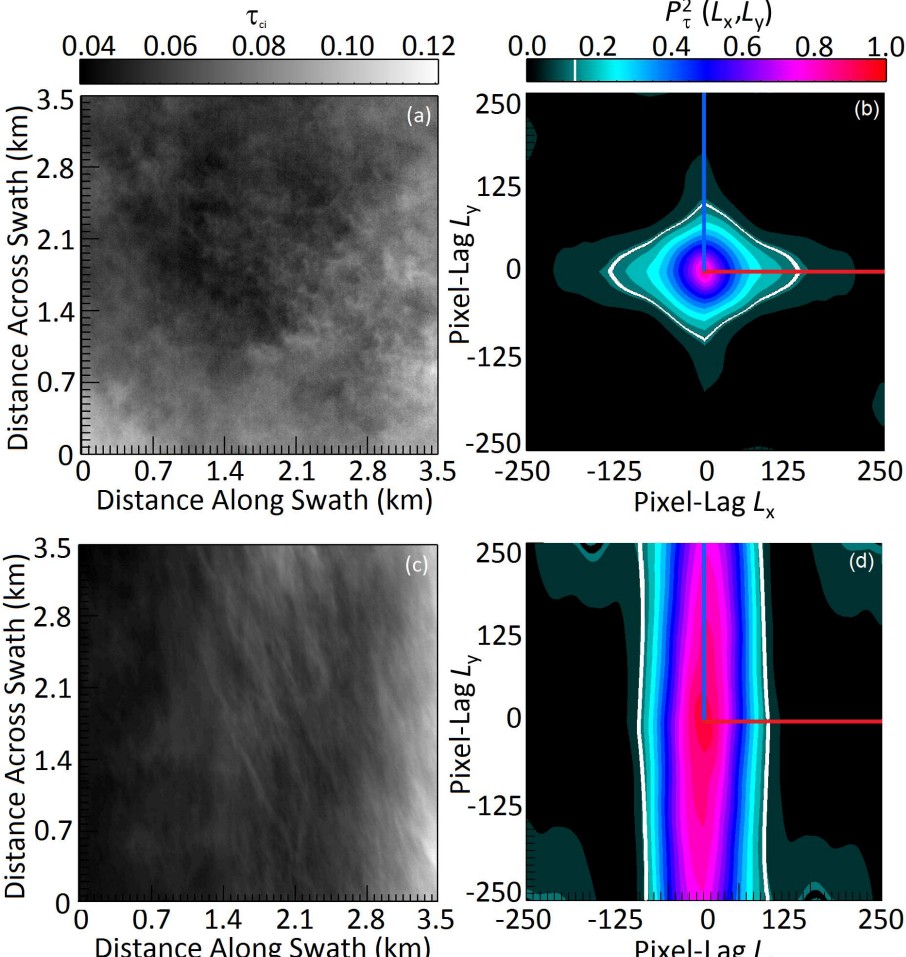

**Figure 2. (a)** Selected cloud scene (3.5 km by 3.5 km) of field of $\tau_{ci}$ from case Ci-01. **(b)** Color-coded 2D field of $P_\tau^2(L_x, L_y)$, calculated for field of $\tau_{ci}$ from (a). The blue and red line illustrate the pixel-lags selected for the illustration in Fig. 3a. The white line illustrates $\xi_\tau$ at $P_\tau^2(L_x, L_y) = 1/e^2$. **(c)** Same as (a) for case Ci-03. **(d)** Same as (b) for selected $\tau_{ci}$ field shown in (c).

Here, $\xi_\tau$ quantifies the length scale (in units of meter) where individual cloud parcels are decorrelated; it provides a measure of the horizontal extent of cloud inhomogeneities. Strong inhomogeneities correspond to small $\xi_\tau$. In Figs. 2b and 2d, $\xi_\tau$ is indicated by a white line. For Ci-01 $\xi_\tau$ forms a circular shape indicating a similar magnitude of cloud inhomogeneities in all directions of the cloud field. Conversely, for Ci-03 $\xi_\tau$ along pixel-lag $L_x$ is significantly smaller than $\xi_\tau$ along pixel-lag $L_y$. This directional dependence is related to the structure of the cloud with regular filaments in the swath direction of the image in Fig. 2c. For case Ci-01 the symmetry in $P_\tau^2(L_x, L_y)$ means that the cloud inhomogeneity can be characterized by a single value $\xi_\tau$, independent of direction. For regularly structured clouds such as Ci-03, however, the 2D decorrelation can be split

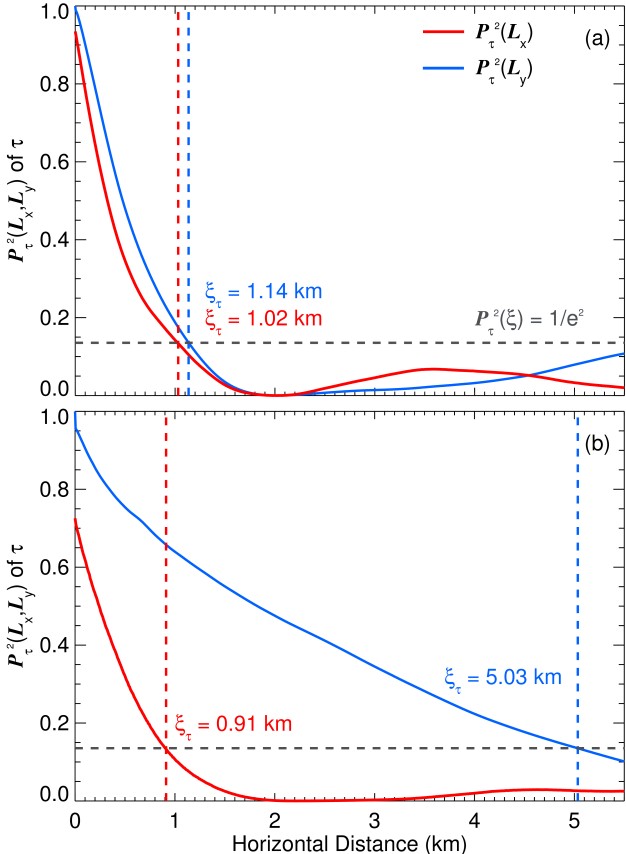

**Figure 3. (a)** Average of squared 1D autocorrelation functions $P_\tau^2(L_x, L_y)$ (solid lines), calculated for pixels, which are orientated into the direction of the blue ($L_x$) and red line ($L_y$) illustrated in Fig. 2b. The dashed lines mark the derived distance of the decorrelation length $\xi_\tau$, where $P_\tau^2(L_x)$ and $P_\tau^2(L_y)$ are decreased to $1/e^2$. **(b)** Same as (a) for $P_\tau^2(L_x, L_y)$ shown in Fig. 2d.

into a component of the largest variability and another one along the smallest variability of $\tau$. In the cloud fields presented here, both major axes align with the $x$ and $y$ direction. 1D autocorrelation functions along the axis of strongest ($\leftrightarrow$, red line in Figs. 2b and 2d) and weakest ($\updownarrow$, blue line in Figs. 2b and 2d) variability are provided in Figs. 3a and 3b. To derive quantitative values for $\xi_\tau^{\leftrightarrow}$ and $\xi_\tau^{\updownarrow}$ in SI-units of meter, the pixel-lag is transformed into horizontal distances by multiplying the

number of pixels by their pixel size.

      For case Ci-01 (Figs. 2a, 2b, and 3a), the derived $P_\tau^2(L_x, L_y)$ along and across the prevailing directional structure are similar and $\bar{\xi}_\tau^{\updownarrow} = 1.14\,\text{km}$ compares well with $\bar{\xi}_\tau^{\leftrightarrow} = 1.02\,\text{km}$ within the range of standard deviation given in Tab. 2. For case Ci-03 (Figs. 2c, 2d, and 3b), the $P_\tau^2(L_x, L_y)$ along and across the prevailing directional structure differ significantly from each other and $\bar{\xi}_\tau^{\updownarrow} = 5.03\,\text{km}$

is about six times larger than $\bar{\xi}_\tau^{\leftrightarrow} = 0.91\,\text{km}$. Thus, for clouds with a prevailing directional structure

it is advisable to give variable $\xi_\tau$ as a function of observation direction, e.g., by two parameters, $\xi_\tau^{\updownarrow}$ along and $\xi_\tau^{\leftrightarrow}$ across the prevailing cloud structure.

The decorrelation lengths are calculated for each measurement case along ($\bar{\xi}_\tau^{\updownarrow}$) and across ($\bar{\xi}_\tau^{\leftrightarrow}$) the prevailing directional structure of the cloud inhomogeneities, which is identified by the 2D au-
tocorrelation analysis. Table 2 summarizes and Fig. 4 illustrates the resulting $\bar{\xi}_\tau^{\updownarrow}$ (blue bars) and $\bar{\xi}_\tau^{\leftrightarrow}$ (red bars). Additionally, $\bar{\xi}_\tau \pm \sigma_\xi$ are included ($\bar{\xi}_\tau^{\updownarrow} \pm \sigma_\xi$, $\bar{\xi}_\tau^{\leftrightarrow} \pm \sigma_\xi$) in Tab. 2. Those values illustrate the pixel-by-pixel variability for the calculated $P_\tau^2(L_x, L_y)$ along one direction. Due to the exponential behaviour of $P_\tau^2(L_x, L_y)$ they are asymmetric with respect to $\bar{\xi}_\tau^{\updownarrow}$ and $\bar{\xi}_\tau^{\leftrightarrow}$.

**Table 2.** Decorrelation length calculated for the retrieved fields of $\tau$ from the CARRIBA (Ci-01–Ci-04) and VERDI (St-01–St-10) campaigns. Vertical arrows ($\updownarrow$) indicate the calculation of $P_\tau^2(L_x, L_y)$ and subsequent derivation of $\xi_\tau$ along $L_y$, horizontal arrows ($\leftrightarrow$) along $L_x$. Furthermore, $\bar{\xi}_\tau$ is the average of all pixels, $\bar{\xi}_\tau - \sigma_\xi$ is the average minus standard deviation, and $\bar{\xi}_\tau + \sigma_\xi$ is the average plus standard deviation.

| Case | $\bar{\xi}_\tau^{\updownarrow} - \sigma_\xi$ (km) | $\bar{\xi}_\tau^{\updownarrow}$ (km) | $\bar{\xi}_\tau^{\updownarrow} + \sigma_\xi$ (km) | $\bar{\xi}_\tau^{\leftrightarrow} - \sigma_\xi$ (km) | $\bar{\xi}_\tau^{\leftrightarrow}$ (km) | $\bar{\xi}_\tau^{\leftrightarrow} + \sigma_\xi$ (km) |
|------|------|------|------|------|------|------|
| Ci-01 | 0.72 | 1.14 | 1.62 | 0.62 | 1.02 | 1.48 |
| Ci-02 | 2.16 | 2.47 | 3.42 | 1.12 | 1.24 | 1.39 |
| Ci-03 | 4.42 | 5.03 | 6.34 | 0.62 | 0.91 | 1.41 |
| Ci-04 | 1.20 | 1.58 | 2.12 | 0.71 | 0.82 | 1.26 |
| St-01 | 0.46 | 0.68 | 0.89 | 0.07 | 0.11 | 0.17 |
| St-02 | 0.67 | 1.12 | 1.56 | 0.11 | 0.18 | 0.26 |
| St-03 | 0.37 | 0.43 | 0.55 | 0.11 | 0.17 | 0.26 |
| St-04 | 0.11 | 0.15 | 0.25 | 0.08 | 0.12 | 0.19 |
| St-05 | 0.13 | 0.17 | 0.26 | 0.07 | 0.10 | 0.17 |
| St-06 | 0.23 | 0.30 | 0.37 | 0.06 | 0.09 | 0.15 |
| St-07 | 0.44 | 0.88 | 1.64 | 0.09 | 0.15 | 0.26 |
| St-08 | 0.30 | 0.73 | 1.57 | 0.13 | 0.16 | 0.24 |
| St-09 | 0.11 | 0.14 | 0.24 | 0.05 | 0.09 | 0.16 |
| St-10 | 0.12 | 0.15 | 0.26 | 0.08 | 0.12 | 0.21 |

The results show that the observed subtropical cirrus yield larger decorrelation lengths $\bar{\xi}_\tau$ than the
Arctic stratus cases. Thus, the subtropical cirrus cases are more homogeneous than the Arctic stratus cases. Furthermore, the results indicate that for most of the measurement cases a distinct directional structure of cloud inhomogeneities is observed. The results for $\bar{\xi}_\tau^{\updownarrow}$ are in 9 of the 14 investigated cases more than twice as large as for $\bar{\xi}_\tau^{\leftrightarrow}$.

For the subtropical cirrus, $\xi_\tau$ varies from $0.82\,\text{km}$ to $5.03\,\text{km}$, depending on the cloud structure
and inhomogeneity. The rather inhomogeneous cases Ci-01 and Ci-04 with highly variable $\tau_{ci}$ on small scales yield rapidly decreasing $P_\tau^2(L_x, L_y)$ with low $\bar{\xi}_\tau$. In contrast, the quite homogeneous

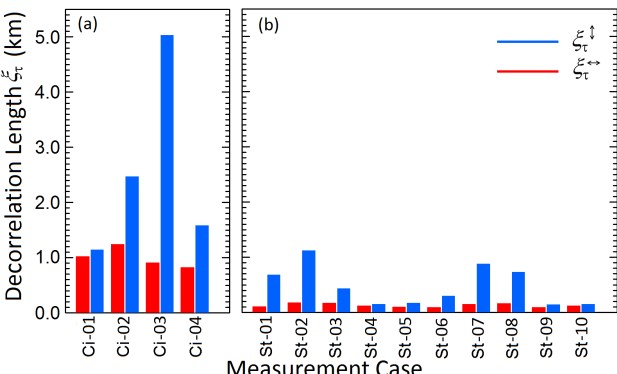

**Figure 4.** Decorrelation length calculated for the retrieved fields of $\tau$ from the **(a)** CARRIBA (Ci-01–Ci-04) and **(b)** VERDI (St-01–St-10) campaigns. Vertical arrows ($\updownarrow$) indicate the calculation of $P_\tau^2(L_\mathrm{x}, L_\mathrm{y})$ and subsequent derivation of $\xi_\tau$ along $L_\mathrm{y}$, horizontal arrows ($\leftrightarrow$) along $L_\mathrm{x}$.

cases Ci-02 and Ci-03 yield slowly decreasing $P_\tau^2(L_\mathrm{x}, L_\mathrm{y})$ and larger $\bar{\xi}_\tau$. The differences between $\bar{\xi}_\tau^{\updownarrow}$ and $\bar{\xi}_\tau^{\leftrightarrow}$ reach up to 82 %.

For the Arctic stratus fields observed during VERDI, $\bar{\xi}_\tau^{\updownarrow}$ and $\bar{\xi}_\tau^{\leftrightarrow}$ range between 0.09 km and
1.12 km. Similar to the CARRIBA cirrus cases, the differences between $\bar{\xi}_\tau^{\updownarrow}$ and $\bar{\xi}_\tau^{\leftrightarrow}$ are significant reaching values of up to 84 %.

However, the absolute values of $\bar{\xi}_\tau^{\updownarrow}$ and $\bar{\xi}_\tau^{\leftrightarrow}$ for the Arctic stratus are smaller (more inhomogeneous) than those for the subtropical cirrus, although the 1D inhomogeneity parameters from Tab. 1 yield similar values. This reveals that the 1D inhomogeneity parameters $\rho_\tau$, $S_\tau$, and $\chi_\tau$ just provide
incomplete information for a comparison of different types of clouds as they are not able to consider the horizontal structure of cloud inhomogeneities. Differences can only be observed by an evaluation of the horizontal pattern of the cloud inhomogeneities.

## 5   Power Spectral Density Analysis

Multiple scattering in inhomogeneous 3D cloud structures causes a smoothing of the reflected ra-
diances $I_\lambda$ above clouds (Cahalan and Snider, 1989; Marshak et al., 1995). This effect generates uncertainties in the retrieved fields of $\tau$ if homogeneous plane-parallel clouds are assumed in the retrieval. Therefore, in this paper the smoothing effect is analyzed using the Fourier transform of the retrieved fields of $\tau$. The application of Fourier transforms for the investigation of cloud inhomogeneities is widely used in the existing literature (e.g., Cahalan, 1994; Davis et al., 1999a; Schröder,
2004). However, in most of these studies, the 1D Fourier transformation is adopted to narrow pixel-lines of radiative quantities such as $I_\lambda$ or the reflectivity $\gamma_\lambda$. Here, a 2D Fourier transformation is applied to spatial 2D cloud scenes. Schäfer et al. (2013) showed that angular features of the scatter-

ing phase functions are imprinted in the $I_\lambda$ measurements of AisaEAGLE. To avoid artifacts in the Fourier transform arising from those features, fields of $\tau$ are used for the analysis.

The Fourier transformation decomposes a periodic function into a sum of sinusoidal base functions. For a given measurement, here $\tau(x,y)$, the 2D Fourier transform $\mathcal{F}_\tau(k_x, k_y)$ is defined by:

$$\mathcal{F}_\tau(k_x, k_y) = \int\limits_{-\infty}^{\infty} \int\limits_{-\infty}^{\infty} \tau(x,y) \cdot e^{-2\pi i \cdot (k_x x + k_y y)} \, \mathrm{d}x \, \mathrm{d}y, \tag{6}$$

The base functions are described by a complex exponential of different frequency. The fields of $\tau$ are
given as a function of horizontal distances $x$ and $y$. Therefore, wave numbers $k_x = 1/x$ and $k_y = 1/y$ are used in the base functions.

The Fourier coefficients $\mathcal{F}_\tau(k_x, k_y)$ are calculated using a Discrete Fourier Transform (DFT). With $n$ and $m$ discrete elements in the $x_j$ and $y_k$ dimension of the $\tau$ field, the 2D DFT is derived by:

$$\mathrm{DFT}(k_x, k_y) = \frac{1}{n \cdot m} \sum_{x_j=0}^{n-1} \sum_{y_k=0}^{m-1} \tau(x_j, y_k) \cdot e^{-2\pi i \cdot \left( \frac{k_x x_j}{n} + \frac{k_y y_k}{m} \right)}. \tag{7}$$

Figures 5 and 6 present the Fourier transform in the form of power spectral densities $E(k_x)$ and $E(k_y)$, in the following called $E(k_x, k_y)$, calculated from the complex Fourier coefficients by:

$$E(k_x, k_y) = \mathrm{DFT}^2(k_x, k_y). \tag{8}$$

    Figures 5a to 5c show $\tau_{ci}$ fields of three selected cloud areas of 3.5 km by 3.5 km size extracted from the cases Ci-01, Ci-02, and Ci-03. Ci-01 represents an inhomogeneous subtropical cirrus with-
out a preferred direction in the cloud structure (Fig. 5a). In Ci-02 a homogeneous subtropical cirrus with a moderate directional structure (Fig. 5b) is selected, while in Ci-03 an inhomogeneous subtropical cirrus with a distinct directional structure (Fig. 5c) is presented. Figures 5d to 5f show the corresponding logarithm of the 2D power spectral densities $E(k_x, k_y)$. Largest values of $E(k_x, k_y)$ are found at smallest wave numbers $k_x$ and $k_y$, which are located in the center of the image. In gen-
eral the values of $E(k_x, k_y)$ decrease with increasing $k_x$ and $k_y$. Inhomogeneous clouds (Figs. 5d and 5f) show higher values of $E(k_x, k_y)$ over a wide range of wave numbers $k_x$ and $k_y$, whereas the dominating $E(k_x, k_y)$ for homogeneous clouds (Fig. 5e) are only located close to the smallest wave numbers $k_x$ and $k_y$. Similar to the autocorrelation functions the decrease of $E(k_x, k_y)$ is rotationally symmetric for clouds with no preferred directional structure (Fig. 5d), but asymmetrical for clouds
with a prevailing directional structure (Figs. 5e, 5f).

    To quantify the two-dimensional nature of the symmetry, Figs. 5g to 5f show the $E(k_x, k_y)$ along (black, $E_b$) and across (red, $E_a$) the direction of the strongest symmetry axis. For the inhomogeneous case without a prevailing directional structure (Ci-01), both components $E_a$ and $E_b$ are almost identical. For the homogeneous case with a moderate directional structure (Ci-02), both $E_a$ and
$E_b$ are similar over most of the covered range of $k_x$ and $k_y$, except for the smallest wave number

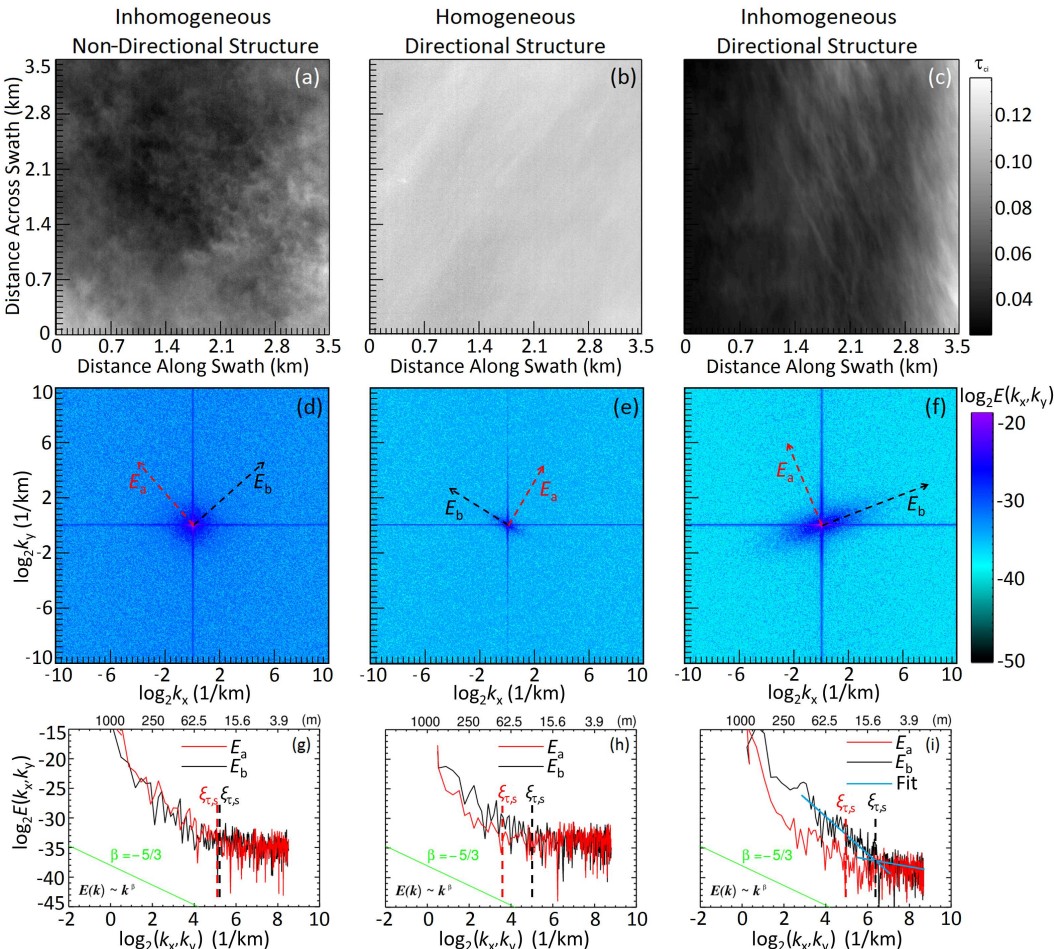

**Figure 5. (a-c)** AisaEAGLE image (3.5 by 3.5 km), **(d-f)** 2D power spectral density $E(k_x, k_y)$, and **(g-i)** 1D power spectral density $E(k_x, k_y)$ across (red arrows, $E_a$) and along (black arrows, $E_b$) the prevailing direction of scale invariant areas for **(a), (d), (g)** inhomogeneous cloud without directional structure, **(b), (e), (h)** homogeneous cloud with slight directional structure, and **(c), (f), (i)** inhomogeneous cloud with distinct directional structure. The $\xi_{\tau,s}$ are marked by colored dashed lines.

$k_x < 3\,\mathrm{km}^{-1}$ and $k_y < 3\,\mathrm{km}^{-1}$. For the inhomogeneous case with a distinct directional structure (Ci-03), both $E_a$ and $E_b$ are of similar magnitude only at $k_x > 7\,\mathrm{km}^{-1}$ and $k_y > 7\,\mathrm{km}^{-1}$. The differences in $E_a$ and $E_b$ of clouds with a prevailing directional structure result from the different $k_x$ and $k_y$, at which the signal turns into white noise (constant $E(k_x, k_y)$, independent of $k_x$ and $k_y$).

350  This transition is used to characterize the small-scale break $\xi_{\tau,s}$, which determines the lower size range of the detected cloud inhomogeneities and identifies the scale at which the measurements turn into white noise. To derive $\xi_{\tau,s}$, fits are applied to the two scale-invariant regimes of $E(k_x, k_y)$ (shown for $E_b$ in Fig. 5i). Subsequently, the small scale break $\xi_{\tau,s}$ is determined as the intersection of those fits. The small scale break $\xi_{\tau,s}$ is connected to the pixel size, which depends on the distance be-

tween cloud and sensor. The corresponding $k_x$ and $k_y$ give $\xi_{\tau,s}$. The small-scale breaks $\xi_{\tau,s}(E_a)$ and

$\xi_{\tau,s}(E_b)$ for case Ci-01 are at about 0.03 km ($\log_2 k_{x,y} \approx 5$). For Case Ci-02, $\xi_{\tau,s}(E_a)$ and $\xi_{\tau,s}(E_b)$

are in the length range of 0.09 km ($\log_2 k_{x,y} \approx 3.5$) and 0.03 km ($\log_2 k_{x,y} \approx 5$), respectively. The

small-scale break $\xi_{\tau,s}(E_a)$ from case Ci-03 is at about 0.03 km ($\log_2 k_{x,y} \approx 5$), while $\xi_{\tau,s}(E_b)$ is at

about 0.01 km ($\log_2 k_{x,y} \approx 6.5$), which is already close to the pixel size, which corresponds to the

lower detection limit that leads to white noise. Thus, $\xi_{\tau,s}$ yields quantitatively larger values along

the prevailing cloud structure than across. Furthermore, the ranges of the derived small-scale breaks

$\xi_{\tau,s}$ are found to be close to the ranges of the small-scale breaks reported in literature. Davis et al.

(1999b) derived small-scale breaks for a broken-stratocumulus/towering cumulus cloud complex

from LWC measurements with a particulate volume monitor probe (4 cm resolution) at ranges of

about 2-5 m. They proposed that those small-scale breaks are related to extreme values in the de-

tected LWC, which appear on small horizontal scales. Besides Poissonian fluctuations of the cloud

optical thickness $\tau$ and the white noise related to power spectral signals at scales below the pixel

size this might be a further explanation for the derived small-scale breaks in the current study and

needs to be investigated in further studies.

Marshak et al. (1995) discussed that cloud inhomogeneity and horizontal photon transport are

scale–dependent processes. The $E(k_x, k_y)$ of cloud optical and microphysical properties are propor-

tional to $k_x^\beta$ and $k_y^\beta$, where $\beta$ is the slope of the power spectral density. At large scales, the $E(k_x, k_y)$

of e.g. $I_\lambda, \tau$, LWC, or IWC follow Kolmogorov's $\beta = -5/3$ law of energy distribution in a turbulent

fluid (Kolmogorov, 1941). At these scales, the variability in the radiation field follows the variabil-

ity in LWC. Increasing cloud inhomogeneity causes a decrease of $\beta$ of optical properties at smaller

scales, but not in $\beta$ of microphysical properties. At scales influenced by horizontal photon transport,

$\beta$ may differ from -5/3 dependent on the cloud inhomogeneity that changes the magnitude of hori-

zontal photon transport. Typically, this affects horizontal scales smaller than 1000 m. The higher the

cloud inhomogeneity, the larger the deviation from -5/3. Thus the slope $\beta$ at scales below 1000 m

provides a measure of cloud inhomogeneity. Usually, the scale break $\xi$ is used to quantify the devi-

ation from -5/3. In the following, the horizontal scale at which the power spectrum starts to deviate

from the -5/3 law defines the large-scale break $\xi_{\tau,L}$. The position of the large-scale break depends

on the size of the horizontal cloud structures; more inhomogeneous clouds with larger variability

on smaller scales yield smaller $\xi_{\tau,L}$. For scales smaller than $\xi_{\tau,L}$, the radiative smoothing leads to

uncertainties in 1D cloud retrievals, where the horizontal photon transport is automatically neglected

(Cahalan, 1994; Marshak et al., 1998; Zinner et al., 2006; Varnai and Marshak, 2007).

A comparison of the $E(k_x, k_y)$ to the -5/3 law in Figs. 5g to 5h shows that the analyzed scenes

are too small to cover the larger scales, which are necessary to identify $\xi_{\tau,L}$. The range of $k_x$ and $k_y$

is lower than $\xi_{\tau,L}$ and $E(k_x, k_y)$ already exhibit a steeper slope than $\beta = -5/3$. Therefore, the size

of the selected areas was extended. Unfortunately, this is only possible for calculations of the DFT

along $L_y$ (across swath). Calculations along $L_x$ (swath) do not cover a sufficiently large distance

to derive quantitative values for $\xi_{\tau,\mathrm{L}}$. Therefore, the following analysis is performed using 1D DFT along $L_{\mathrm{y}}$ only. Furthermore, both cloud cases, subtropical cirrus and Arctic stratus, exhibit a similar pixel length along $L_{\mathrm{y}}$ (5±2 m), which results from the chosen frame rate (subtropical cirrus: 4 Hz, Arctic stratus: 30 Hz) and given cloud ($\approx 20\,\mathrm{m\,s^{-1}}$) and aircraft ($\approx 70\,\mathrm{m\,s^{-1}}$) velocity. This allows a direct comparison between these two different cloud types with different observation geometry.

Figures 6a and 6b show the 1D DFT calculated across the swath for two typical cases of subtropical cirrus (CARRIBA case Ci-01) and Arctic stratus (VERDI case St-07). The two cases are selected, since they exhibit a similar length $L_{\mathrm{y}}$. For each line of the $\tau$ field (each swath pixel) $E(k_{\mathrm{y}})$ is calculated and the individual power spectra are overlaid as gray dots in Fig.6. To evaluate the resulting 1D Fourier spectra with reduced noise (rn) characteristics, the $E_{\mathrm{rn}}(k) \sim k^{\beta}$ are calculated with the use of octave binning, following the method proposed by Davis et al. (1996), Harris et al. (1997), and Schröder (2004). Logarithmically spaced bins $k_{\mathrm{n}}$ are calculated by:

$$k_n = \frac{1}{2^n} \sum_{i=2n}^{2^{n+1}-1} k_i, n = 1, 2, ..., \log_2(N-2),\tag{9}$$

for the number of data points $N$. $E(k_{\mathrm{rn}})$ is then obtained by:

$$E(k_{\mathrm{rn}}) = \frac{1}{2^n} \sum_{i=2n}^{2^{n+1}-1} E(k_i), n = 1, 2, ..., \log_2(N-2).\tag{10}$$

Within each bin $2^n$ data points are averaged. In addition to the reduced noise of $E(k_{\mathrm{rn}})$ compared to $E(k_{\mathrm{y}})$ the binning provides a uniform contribution of all scales to the average values.

The $E(k_{\mathrm{rn}})$ derived from the octave binning are included as green diamonds in Fig.6. The data of the octave binning were used to fit the spectra for different slopes in the different scale ranges. A green line indicates the $\beta = -5/3$ law. For large scales, the $E_{\mathrm{rn}}(k_{\mathrm{y}})$ (blue fit) approximately follow the $-5/3$ relation in both cases. The large-scale break ($\xi_{\tau,\mathrm{L}}$) is evident at the intersection between the blue and the red line. Here, the slope in the $E_{\mathrm{rn}}(k_{\mathrm{y}})$ becomes steeper. For the CARRIBA case $\xi_{\tau,\mathrm{L}} = 0.31$ km and the middle scale slope $\beta_{\mathrm{m}}$ decreases to -2.2. For the VERDI case $\xi_{\tau,\mathrm{L}} = 0.06$ km and $\beta_{\mathrm{m}}$ decreases to -2.2. The middle–scale slope $\beta_{\mathrm{m}}$ is a function of the inhomogeneity in the measured signals. With increasing inhomogeneity of the optical thickness $\tau$, $\beta_{\mathrm{m}}$ decreases. Together with the smaller $\xi_{\tau,\mathrm{L}}$, this indicates that the selected Arctic stratus case is more inhomogeneous compared to the selected subtropical cirrus case. As discussed above, $\xi_{\tau,\mathrm{s}}$ is observed at the intersection between the fits for the middle (red, $\beta_{\mathrm{m}}$) and small scales (orange, $\beta_{\mathrm{s}}$). Due to the analysis of a significant larger distance compared to Fig. 5, it is highly uncertain to give quantitative numbers for $\xi_{\tau,\mathrm{s}}$. Therefore, it is indicated only qualitatively. However, $\xi_{\tau,\mathrm{s}}$ identifies at which scales the measurements turn into noise. The scale depends on the distance between sensor and cloud. For the sensor, noise dominated at scales two times pixel range, which corresponds to about 15 m for the subtropical cirrus observations ($\approx 12$ km cloud base altitude) and 3.5 m for the Arctic stratus observations where the aircraft was closer to cloud top ($\approx 2$ km distance).

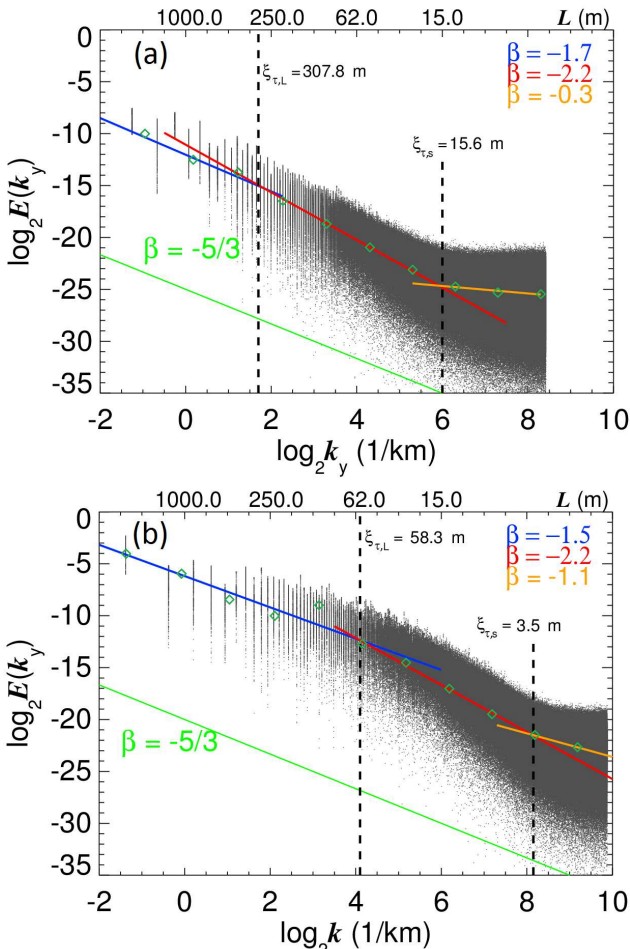

**Figure 6.** 1D power spectral density $E(k_y)$ (gray dots) for each spatial pixel on the swath axis of the $\tau$ field from **(a)** case Ci-01 and **(b)** case St-07. Scale-invariant slopes $\beta$ are marked with colored solid lines. The $E(k_{rn})$ derived from the octave binning are included as dark green diamonds. Scale breaks $\xi_{\tau,L}$ and $\xi_{\tau,s}$ are indicated by dashed lines.

Figure 7 illustrates $\xi_{\tau,L}$ for all available cloud cases from CARRIBA and VERDI. Especially the values for the Arctic stratus are in the size range, which was also reported by Marshak et al. (1995), who found scale breaks for fractal clouds in the range of 200–500 m. Furthermore, the values compare well with the derived decorrelation length $\xi_{\tau}$ derived in Sect. 4. Although the exact values 430 of $\xi_{\tau,L}$ are not equal to $\xi_{\tau}$, both are in the same size range for each individual case. Similar to $\xi_{\tau}$ given in Tab. 2, $\xi_{\tau,L}$ confirms that Ci-02 and Ci-03 are more homogeneous than Ci-01 and Ci-04. Furthermore, the resulting large scale breaks $\xi_{\tau,L}$ confirm the results from the derived decorrelation lengths $\xi_{\tau}$ that the subtropical cirrus observed during CARRIBA is more homogeneous (larger $\xi_{\tau}$ and $\xi_{\tau,L}$) than the Arctic stratus from VERDI (smaller $\xi_{\tau}$ and $\xi_{\tau,L}$). This is related to the fact that

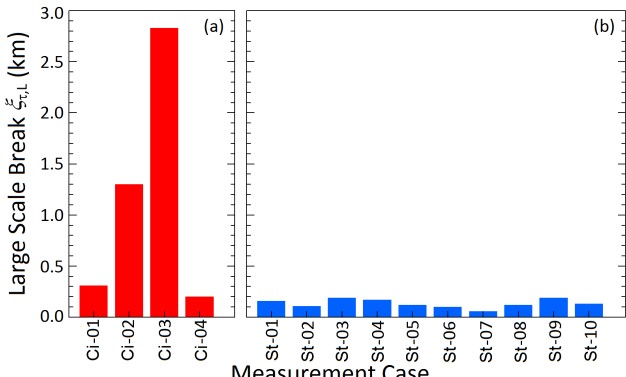

**Figure 7.** Across swath derived large scale breaks $\xi_{\tau,\mathrm{L}}$ for the retrieved fields of $\tau$ from the **(a)** CARRIBA (Ci-01 to Ci-04, red) and **(b)** VERDI (St-01 to St-10, blue) campaigns. The values were derived by using the method presented in Fig. 6.

$\xi_{\tau,\mathrm{L}}$, which is the radiative smoothing scale, is a function of the cloud geometrical thickness and the transport mean free path. For Arctic stratus both parameters are significantly smaller than for subtropical cirrus.

An estimation of the uncertainty in the derived $\xi_{\tau,\mathrm{L}}$ can be obtained from a comparison to investigations performed by Schröder and Bennartz (2003). Amongst others, Schröder and Bennartz (2003) investigated scale breaks as a function of wavelengths and absorption bands. Their results show uncertainties in a range of 3 % to 8 %. Schröder and Bennartz (2003) derived those uncertainty values by subsetting the points of the power spectrum that are used for the slope fit. Using this method, they obtained a set of different slopes and scale breaks. The particular standard deviations of those sets are used as an uncertainty for the octave binning method. Applied to the VERDI cases, the 3 % to 8 % result in a maximum uncertainty of $\pm\,5\,\mathrm{m}$ (St-07) to $\pm\,15\,\mathrm{m}$ (St-03, St-09) in the derived $\xi_{\tau,\mathrm{L}}$. For the CARRIBA cases the maximum uncertainty is in the range of 16 (Ci-04) to 226 m (Ci-03). However, especially case Ci-03 is characterized as rather homogeneous. Therefore, much lower uncertainty values are to be expected.

## 6  Summary and Conclusions

During the two field campaigns CARRIBA and VERDI, downward (ground-based) and upward (measured from aircraft) fields of solar spectral radiance ($I_\lambda^\downarrow$, $I_\lambda^\uparrow$) were measured with high spatial resolution (less than $10\,\mathrm{m}$), using the imaging spectrometer AisaEAGLE. The measured radiance fields were used to retrieve fields of $\tau$, which were subsequently analyzed to quantify horizontal cloud inhomogeneities. Furthermore, due to the observation of 2D fields, the prevailing directional structure of the cloud inhomogeneities was investigated.

Four subtropical cirrus cases collected during CARRIBA and ten Arctic stratus cases sampled during VERDI were studied in detail. The cloud inhomogeneity was quantified by three 1D inhomogeneity parameters $\rho_\tau$, $S_\tau$, and $\chi_\tau$, as well as 1D and 2D autocorrelation functions, and Fourier analysis.

Considering the pixel and domain size of the analyzed measurements, the results from the calculated 1D inhomogeneity parameters $\rho_\tau$ and $S_\tau$ are in agreement with values given in the literature for similar cloud types. The calculated $\rho_\tau$ are in the range of 0.17–0.91 for the subtropical cirrus observed during CARRIBA and 0.15–0.34 for the Arctic stratus measured during VERDI. The literature values are in the range of 0.07–0.78. The inhomogeneity parameter $S_\tau$ exhibits values of
0.08 to 0.48 for CARRIBA and 0.07 to 0.20 for VERDI, which agrees with values of 0.03 to 0.3 given in literature. For $\chi_\tau$, the literature estimates values between $\approx 0.65$ and 0.8, while the results from CARRIBA and VERDI are significantly larger. This is probably related to the different pixel and domain sizes. All values except for Ci-04 ($\chi_\tau = 0.63$) are in the range between 0.92 and 0.99. A further comparison between the results for the clouds encountered during CARRIBA and
VERDI showed that all three 1D inhomogeneity parameters exhibit values of similar magnitude for both cloud types; subtropical cirrus and Arctic stratus. This might lead to the conclusion that the inhomogeneity of both cloud types could be treated by the same 1D inhomogeneity parameters.

However, the comparison of the 2D analysis of squared autocorrelation functions $P_\tau^2(L_x, L_y)$ with the 1D inhomogeneity parameters $\rho_\tau$, $S_\tau$, and $\chi_\tau$ showed that it is important to consider the full
horizontal structure of clouds using 2D analysis rather than 1D analysis, when determining cloud inhomogeneity. For both cloud cases (subtropical cirrus, Arctic stratus) the 1D inhomogeneity parameters yield similar values, but significant differences resulting from the analysis of $P_\tau^2(L_x, L_y)$, which additionally contain information about the horizontal structure of cloud inhomogeneities. The 1D inhomogeneity parameters are not capable of differentiating the directional structure of clouds
and may lead to misinterpretations of cloud inhomogeneity. From the squared autocorrelation functions $P_\tau^2(L_x, L_y)$ the decorrelation length $\xi_\tau$ was derived, which is a measure of the size range of the cloud inhomogeneities. The 2D analysis of $P_\tau^2(L_x, L_y)$ revealed that $\xi_\tau$ is a function of the directional structure of the cloud inhomogeneities. Without knowledge of the directional structure of cloud inhomogeneities, no universally valid value for $\xi_\tau$ can be derived from the retrieved fields of
$\tau$. The differences in $\xi_\tau$ as derived from a 1D autocorrelation analysis along and across the prevailing structure of cloud inhomogeneities reached up to 82 % and 84 % for CARRIBA and VERDI, respectively. It is concluded that the directional cloud structure has to be taken into account for a quantification of cloud inhomogeneities. The absolute values of $\xi_\tau$ were in the range of 0.82 km to 5.03 km for CARRIBA and 0.09 km to 1.12 km for VERDI. Furthermore, the results from the
2D analysis showed that for the observed cloud cases the subtropical cirrus was more homogeneous than the Arctic stratus. This result was not available from the investigation of the commonly used 1D inhomogeneity parameters. Therefore, using 2D methods in future studies for the characterization

of cloud inhomogeneities is advisable, since their information content exceeds the information content of the commonly used 1D inhomogeneity parameters. Nowadays, 2D images of cloud fields are widespread by e.g., measurements of all-sky cameras or satellite observation with high spatial resolution. Applying the presented methods to such continuous measurements would provide detailed views into the climatology of cloud inhomogeneities.

3D radiative effects are quantified by applying 2D Fourier transformation to the retrieved fields of $\tau$. The power spectral densities $E(k_{x,y})$ calculated from the Fourier Transform of $I_\lambda^\downarrow$ and $I_\lambda^\uparrow$ show evidence that 3D radiative effects did affect the radiation field of both cloud types, subtropical cirrus and Arctic stratus. For larger scales ($> 1000\,\text{m}$), no horizontal photon transport was observed as the $E(k_{x,y})$ followed Kolmogorov's -5/3 law. Approaching smaller scales ($< 1000\,\text{m}$), the derived slopes become steeper indicating radiative smoothing by cloud inhomogeneities and horizontal photon transport. From the intersection of fits of the three slope regimes, the small-scale break $\xi_{\tau,s}$ (between small- and middle-scale slopes) and the large-scale break $\xi_{\tau,L}$ (between middle- and large-scale slopes) were derived. Similarly to the analysis using autocorrelation functions, $\xi_{\tau,s}$ depends on the directional structure of the cloud inhomogeneities. Due to a too small swath width, a similar analysis for $\xi_{\tau,L}$ could not be performed. However, the calculated $\xi_{\tau,L}$ along the image are comparable to the results derived from the analysis of $P_\tau(L_x, L_y)$. The large-scale break $\xi_{\tau,L}$ for CARRIBA was in the range of $0.20\,\text{km}$ to $2.83\,\text{km}$. For VERDI a range of $0.06\,\text{km}$ to $0.19\,\text{km}$ was covered by $\xi_{\tau,L}$.

In early studies, by e.g. Marshak et al. (1998) or Schröder (2004), the scale dependence of cloud radiation measurements was analyzed along one direction (narrow pixel-lines) using 1D DFT. However, the resulting $E(k)$ are valid for the particular observation direction along the given path only. Due to prevailing wind directions, clouds tend to evolve directional structures. In such cases, the calculated $E(k)$, $\beta$, $\xi_{\tau,s}$, and $\xi_{\tau,L}$ will only be valid for the whole cloud if the cloud structure exhibits a non-directional character (compare Figs. 2b and 3a). In all other cases, significant differences can be expected (compare Figs. 2d and 3b). We found such differences for more than the half of the observed cloud scenes. Therefore, the directional structure of cloud inhomogeneities should be taken into account, when cloud inhomogeneities are characterized. It is expected that the information content derived from the directional analysis of cloud inhomogeneities can clearly improve sub-grid scale parametrizations in weather and climate models. For this, depending on the application, the decorrelation length (size and structure of cloud inhomogeneities) or the scale breaks (horizontal photon transport, 3D radiative effects) may provide better proxies compared to commonly used 1D inhomogeneity parameters.

However, so far only two cloud types were investigated. To build up a better idea on cloud inhomogeneity of different cloud types, more high definition observations of cloud fields are needed. Beside dedicated field campaigns, continuous observations by all-sky cameras or satellites with high

spatial resolution such as LandSat (15-90 m resolution) or ASTER (Advanced Spaceborne Thermal Emission and Reflection Radiometer, 15-90 m resolution) may provide the required data.

The 1D and 2D autocorrelation functions and Fourier analysis in conjunction with the derived decorrelation length and scale break are a helpful tool to verify cloud resolving models in terms of typical horizontal cloud geometries.

*Acknowledgements.* This study was supported by the German Research Foundation (Deutsche Forschungsgemeinschaft, DFG) as part of the CARRIBA project (WE 1900/18-1 and SI 1534/3-1). We gratefully acknowledge the support by the SFB/TR 172 "ArctiC Amplification: Climate Relevant Atmospheric and SurfaCe Processes, and Feedback Mechanisms (AC)[3]" in Project B03 funded by the DFG. We thank the Max Planck Institute for Meteorology, Hamburg for supporting the ground-based radiation measurements with the infrastructure of the Barbados Cloud Observatory at Deebles Point on Barbados. We are grateful to the Alfred Wegener Institute Helmholtz Centre for Polar and Marine Research, Bremerhaven, Germany for supporting the VERDI campaign with the aircraft and manpower. In addition we like to thank Kenn Borek Air Ltd., Calgary, Canada for the great pilots who made the complicated measurements possible. For excellent ground support with offices and accommodations during the campaign we are grateful to the Aurora Research Institute, Inuvik, Canada.

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
