# Peer review of "Directional, Horizontal Inhomogeneities of Cloud Optical Thickness Fields Retrieved from Ground-Based and Airborne Spectral Imaging"

_Atmospheric Chemistry and Physics, 2016_

## Referee Comment (RC1) · Anonymous Referee #1 · 20 Sep 2016

The paper uses established analysis techniques to examine inhomogeneity characteristics of two different types of cloud systems at different locations measured by radiometers that were either ground-based or aboard an aircraft. Two aspects of cloud inhomogeneity are examined: inhomogeneity that arises solely by the nature of the PDF of cloud optical thickness, i.e., inhomogeneity from one-point statistics that do not capture the spatial arrangement/coherence of individual cloud elements, and inhomogeneity that depends on the exact nature of the spatial arrangements. The study does not really provide new insights, but rather focuses on the information content of available radiometric cloud measurements and how much diversity exists in the observations (different days for the same general cloud type, different cloud types).

Comments:

– Lines 46-60. A very incoherent paragraph. Cloud inhomogeneity effects on gridded fluxes are mixed with effects on satellite retrievals. Lines 49-50 talk about retrievals, and the next sentence talks about GCMs. Moreover, the first problem facing GCMs is not the lack horizontal photon transport, but the absence of subgrid variability, i.e., the unavailability of PDFs of cloud condensate for each layer. If such PDFs were available at least IPA calculations would in principle be possible (still no horizontal transport). The dissemination of confusion continues later on. In lines 55-58 the limitations of IPA compared to 3D are followed in lines 58-60 by an irrelevant example of errors found by Shonk and Hogan when comparing PPH and IPA.

– Lines 61-68. Here, cloud overlap is mixed into 3D effects and Monte Carlo discussion. Misleading. You can account for overlap perfectly, but still ignore 3D effects by performing IPA calculations on the perfectly overlapped cloud field.

– Lines 75-77. I'm sure that ECMWF models do not need the two point statistics of cloud structure (as derived by autocorrelation and power spectrum analysis), but some information on the PDF, i.e., the inhomogeneity parameters of section 3 derived from one point statistics, so invoking "spatial features" "below the meter scale" is again inappropriate.

– Lines 141-142: "However, the fact that rho_tau can exceed values of unity and depends on the average value might lead to misinterpretations." Why? I don't see anything wrong with values greater than unity.

– Lines 157-158: It would be simpler to say that chi is the scaling factor with which mean tau needs to be multiplied to approximate the IPA albedo.

– Page 7: One has to be careful when comparing inhomogeneity parameters across publications. First, pixel size matters. Second, and most importantly, the domain size matters. The bigger your reference domain, the wider the PDF, the larger the inhomogeneity. So this is by no means a trivial comparison.

– Lines 188-190. It's not a matter of directional dependence only. It's mostly a matter of spatial coherence of cloud condensate, in other words how the variability is distributed across scales.

– Lines 204-205: Why are negative autocorrelations ambiguous?

– Eq. (5): If the scale length is typically defined as the distance at which the autocorrelation drops to 1/e, shouldn't the scale length of squared autocorrelation be defined as the distance where it drops to $1/e^2$?

– Lines 255-256: It's not that they are not well-suited, it's that by themselves they provide incomplete information, i.e., not the whole story.

– Power spectrum scale break analysis: Have the authors given any thought on whether the comparisons of scales in terms of physical units (m) makes sense when the pixel sizes are different? With pixel size varying, the extent to which the smoothing is resolved is also different, so I was wondering whether defining the scale lengths in terms of pixel number would bring the results closer together.

Typos and other minor stuff: – Line 10: "VERtical" instead of "VERical". Also Line 97.

– Lines 120 and 272: I think you wanted to use "fingerprint' rather than "footprint". The term "footprint" in remote sensing indicates the resolution, i.e., pixel size.

– Line 328: "inhomogeneity".

– Line 337: "too small".

---

## Referee Comment (RC2) · Anonymous Referee #2 · 11 Oct 2016

**Review of M. Schäfer, et al. paper "Directional, Horizontal Inhomogeneities of Cloud Optical Thickness Fields Retrieved from Ground-Based and Airborne Spectral Imaging" submitted to ACP**

The paper provides a thorough analysis of cloud inhomogeneity. It also provides a fresh look on many approaches developed earlier. It is easy to read and follow. I enjoyed reading it as many other papers by this Leipzig group that I used to follow earlier. The paper definitely deserves to be published in ACP after some revision and I'm sure that it will be well cited.

However, in addition to the comments and suggestions listed below, my main concern with the paper is the lack of a clear statement on what new we were supposed to learn at each step of both (the one-point and the two-points) analyses provided in the paper. What is the main message the authors want us to take home after reading it? I got a feeling that the paper is much more descriptive than conclusive. I'd like to see a list of bullets/statements, at least, in the 'Summary and Conclusion' section.

**The list of references** is very rich but, as always, is incomplete. I would definitely add two more very relevant papers. The first one is

*Davis, A., Marshak, A., Gerber, H., and Wiscombe, W., 1999: Horizontal structure of marine boundary-layer clouds from cm– to km–scales. J. Geophys. Res. 104, 6123-6144.*

In this paper the authors discuss the structure of marine stratocumulus clouds down to 4-cm scale using both spectral and structure function analyses. Another paper is

*Barker, H.W., B. A.Wielicki, and L. Parker, 1996: A parameterization for computing grid-averaged solar fluxes for inhomogeneous marine boundary layer clouds. Part II: Validation using satellite data. J. Atmos. Sci., 53, 2304–2316.*

(May be also the Part 1). This paper, I believe, was the first to use the ratio $\nu=(<\tau>/\sigma_\tau)^2$ to quantify cloud inhomogeneity.

**Small-scale break $\xi_{\tau,S}$.** I wonder if the small-scale noise can be reduced by averaging over all cases or over all columns (or rows) in one case. Also, please, compare the location of your small-scale breaks with the once reported by Davis et al. (1999). It was not clear for me what could be learned about cloud structure from the reported small-scale breaks. How does your conclusion depend on pixel size and uncertainty in observations? Please summarize.

**Large-scale break $\xi_{\tau,L}$.** After Fig. 7, I'd recommend to mention that

$$\text{CARRIBA } \xi_{\tau,L} << \text{VERDI } \xi_{\tau,L}$$

especially, for the most homogeneous cases of C-02 and C-03. This is partly because $\xi_{\tau,L}$, as the radiative smoothing scale, is the harmonic mean of the cloud geometrical thickness and the transport mean free path. Both factors are much smaller for

Arctic stratus than for cirrus.  I'd also recommend comparing the theoretical values of the radiative smoothing scale with the observed ones, $\xi_{\tau,L}$.

**Retrieval of $\tau$**.  I know that several references on the retrieval processes are given. However, the way $\tau$-field has been retrieved is important for understanding the analysis provided in the paper.  The main question is how much the retrieved $\tau$-field is influenced by 3D radiative effects.  I'd recommend to briefly describing here the retrieval processes.  Another point, I was not convinced that from analyzing the structure of the retrieved cloud optical depth fields for inhomogeneity, one can learn something new compared to the analysis of the measured fields of radiance. I'd recommend, in parallel to, say, Figs. 5 or 6 (or even 7), showing some results of the analysis of energy spectra for the radiance fields.

**Decorrelation length $\xi_{\tau}$**.  I wonder why did you use $1/e$ for the squared autocorrelation function rather than $1/e^2$.

---

## Referee Comment (RC3) · Anonymous Referee #3 · 19 Oct 2016

This study compares a number of different metrics that describe horizontal cloud inhomogeneity. These metrics all derive from different papers, although the authors here calculate them all for the same set of cloud fields, which are derived from ground-based and aircraft measurements of cirrus and stratus from two field campaigns. The authors start with the traditional one-dimensional metrics, but then evaluate metrics that describe the clouds in two dimensions, which allow for differences in inhomogeneity parameter that is due to different structures, for example, along-wind and across-wind.

The paper is generally well-structured and the figures clearly lead the reader through the material. Some parts of the text, however, are not easy to follow and need revising – see the various comments below. However, and this is perhaps my greatest comment

on this paper, there is no real context to the work, no stated reason why the study is being carried out, and at the end there is no discussion of the implications of this study on the scientific community. More context and discussion needs to be added before the paper is ready for publication.

Comments. . .

Abstract – there are too many technical terms in the abstract. While reading it, it was not clear to me what the "decorrelation lengths" and "scale breaks" refer to (lines 11 to 14), and without further context, I did not know what the authors meant by "directional cloud inhomogeneities". All became clear having read the paper, although this defeats the point of the abstract – it should be a standalone block of text that is entirely contained. These terms need clarification in the abstract.

Section 1 – the authors cover a great deal of literature in this section, although in places in a somewhat haphazard manner – they may want to have another think about the structure of the section. See following comments.

Lines 41 to 42 – to me it is unclear what the sentence "a variability of cirrus albedo of up to 25% due to spatial cirrus inhomogeneity" means. I assume it refers to a plane-parallel bias, although the sentence appears in advance of the discussion of the plane-parallel approximation (from line 46).

Lines 46 to 60 – this paragraph is very muddly and confusing. It starts off introducing the plane-parallel approximation, before jumping to the independent pixel approximation, which they say is used in 1D radiative transfer calculations, but it follows after a mention of GCMs, perhaps suggesting that the IPA is used in GCMs. This paragraph should be reworded such that the distinction is clearer. The paragraph continues to cite 50% and 8% errors from two different studies, although no comment is made as to why these are so different.

Lines 64 to 68 – there is some confusion in this paragraph. It starts with a mention of

Monte-Carlo simulations, and then suddenly into cloud overlap schemes, which are a completely different area to horizontal cloud inhomogeneity. The authors also assert that Tripleclouds is a cloud overlap scheme – this is not true: it is a method for representing horizontal cloud inhomogeneity in GCMs. Additionally, the scheme is called "Tripleclouds", not "Triplecloud". The following description of Huang and Liu's (2014) method is also not clear. This part of the literature review may need restructuring – as the paper is dealing with metrics of horizontal inhomogeneity, it would be better to introduce methods of accounting for horizontal inhomogeneity in a more clear manner – plus the authors may want to include the Monte-Carlo ICA method of Pincus et al (2003).

Line 75 – why has the ECMWF mode been singled out as an example here? Surely all models need sub-gridscale parameterisations of cloud structure.

Line 84 – the authors say that 1D measures of inhomogeneity can lead to unrealistic results. To describe them as unrealistic does not really feel fair. A 2D metric of horizontal cloud structure is more descriptive of the cloud field, but 1D metrics can still provide information about the "bulk" structure of the cloud field over all angles.

Lines 90 to 94 – the entire scope and contents of the paper are summed up into a short paragraph of four lines. The contents of the paper could be described in a little more detail. Also, this paragraph (and indeed section 1) is lacking in aims and motivations of the study – indeed, all it says is what the authors will do in their paper, with no science questions. This needs expanding.

Lines 99 to 100 – this is not a particularly important point, but it would be great to get a better idea of how the scans are made – for example, are they made from a long time series of 1D scans across a path, or are they full 2D scans that overlap? The papers referred to here probably describe this fully, but it would be nice to have a little more information here.

Lines 115 to 121 – the wording of this argument is not particularly clear. I think the

authors are saying that optical depth is better to use for this analysis because it is not dependent on scattering angle, while radiance is more directional. This could be better explained.

Line 142 – the authors say that the normalised inhomogeneity parameter can exceed one in some cases, and that this could lead to misinterpretation. I am not sure what they mean by misinterpretation – mathematically, it can happen.

Page 7 – through much of this page, the authors compare the different measures of 1D inhomogeneity from their study with values from other studies and say that they "compare well" and similar. It should be noted that these inhomogeneity parameters can be highly related to the domain and pixel size of the observations in the studies (see Shonk et al 2010, part one). To say that values "compare well" is probably only fair to use when comparing with studies that use very similar domain/pixel sizes. The same is also true in the conclusions in lines 398 to 399.

Line 204 – the authors take the square of the autocorrelations to get an absolute correlation, which is fine, although I would contend that finding negative correlations is not necessarily ambiguous.

Line 209 – if the positive and negative signs indicate the shifting direction, I would say they do have physical meaning.

Figure 3 – this caption could do with a little more information. The caption is entirely written in mathematical terms – it is clear what these mean by reading the text, but defining them briefly in the caption could be helpful.

Figure 4 – why are the red and blue bars stacked? This gives the impression that the authors are summing together the $L_x$-direction and $L_y$-direction decorrelation length scales – by the looks of Table 2, this is not the case. I assume they have been arranged in this way because the blue bar is always taller, implying that the decorrelation length in the $L_y$-direction is always greater. (Which itself is an interesting question that is not

[Figure]

addressed.)

Lines 256 to 257 – the authors provide a conclusion for section 4 with the sentence "A comparison can only indicate . . . inhomogeneities." Many aspects of this sentence are unclear – indeed, I am not sure what the authors are concluding.

Lines 348 to 352 – the authors introduce the reduced noise terms, but again perhaps a little too technically for someone who does not know how to do it – they may want to consider revising these few lines to make the method clearer.

Line 375 – is the CARRIBA cirrus "more homogeneous" than the VERDI stratus? Earlier, the authors say that cases C-02 and C-03 are more homogeneous than C-01 and C-04, and in Table 2, we see that they have larger decorrelation length scales. But the decorrelation scales from the VERDI stratus are smaller than those from the CARRIBA cirrus, which surely means that the cirrus are more inhomogeneous than the stratus? Or am I missing something?

Line 408 – I think this first sentence is saying that, when determining cloud inhomogeneity, it is important to consider the full horizontal structure of clouds via a 2D metric rather than a 1D metric. The authors may want to revisit this sentence – at the moment, it seems to imply that performing 2D analysis shows that 1D analysis works.

Lines 443 to 444 – the final conclusion seems to be that, despite having gone to all the trouble of calculating several different measures of 1D and 2D inhomogeneity, "the directional structure of cloud inhomogeneities generally should be taken into account". This feels like a weak concluding comment. For a start, what is meant by "generally"? Clearly some clouds will have less directional dependence of cloud inhomogeneity, but this does not mean it is not useful to measure it. How is best to determine this two-dimensional inhomogeneity? Which of the methods compared performed best? Which is easiest to calculate? Which is most useful if, say, 2D inhomogeneity parameters were to be included in a climate model? Also, how easy is it to obtain observational cloud data to extract these 2D cloud fields – are they readily available, or should we

be carrying out more of these field campaigns to build up a better idea of cloud inhomogeneity? And is there any cause to investigate 3D cloud inhomogeneity metrics? These are a few questions that popped up in my mind having read the paper – it is disappointing that the authors have not included any such consideration on the implications of these results on the scientific community, and how the work could be taken forward.

Specific comments...

Line 9 and 97 – should "VERical" read "VERtical" (or is it spelt like that in the project name)?

Line 105 – "...full width at...", not "...full with at...".

Table 1 – the four cirrus cases are all labelled "C-01" – should these be "C-01", "C-02", and so on?

Line 192 – "both dimensions", not "both dimension".

Figure 4 – "Decorrelation" on the y-axis has been written as "De-Correlation".

Line 239 and 247 – the value of 3154 m is shown as 3153 m in Figure 3.

Line 262 – "in a horizontal direction", not "in horizontal direction".

Lines 304 to 305 – "$E_a$ and $E_b$" has been written as "$E_a$ and $E_a$".

Lines 328 and 329 – "inhomogeneity" has been written twice as "inhomogeniety".

Line 337 – "too small", not "to small".

Line 346 – "overlaid", not "overlayed".

Line 360 – "decreases to 2.2" should be "decreases to –2.2" (based on the numbers on Figure 6).

Figure 7 – the caption seems to be a copy of the caption of Figure 6.

[Figure]

Line 408, 411 and 435 – the authors switch here between $P_\tau$ and $P_\tau^2$ – is there any reason for this?

Line 473 – "Green-function", not "green-function" (I think they are named after someone called Green).

Line 480 – "absorption", not "absorbtion".

Lines 524 and 526 – check the spelling of "Schroeder" vs "Schröeder".

Line 538 – "Earth's", not "Earths".

Finally, the authors have started several sentences throughout the paper with numbers and variables – this is something that may get picked up later in the review process, but they may want to revise these sentences so that they start with words.

---

## Author Comment (AC1) · 15 Dec 2016

We thank the reviewer for the helpful comments which improved the manuscript significantly. The detailed replies on the reviewer's comments are given below and structured as follows. Reviewer comments have bold letters, are labeled, and listed always in the beginning of each answer followed by the author's comments including (if necessary) revised parts of the paper. The revised parts of the paper are written in quotation marks and italic letters.

**Comments:**

1. **Lines 46-60. A very incoherent paragraph. Cloud inhomogeneity effects on gridded fluxes are mixed with effects on satellite retrievals. Lines 49-50 talk about retrievals, and the next sentence talks about GCMs. Moreover, the first problem facing GCMs is not the lack horizontal photon transport, but the absence of subgrid variability, i.e., the unavailability of PDFs of cloud condensate for each layer. If such PDFs were available at least IPA calculations would in principle be possible (still no horizontal transport). The dissemination of confusion continues later on. In lines 55-58 the limitations of IPA compared to 3D are followed in lines 58-60 by an irrelevant example of errors found by Shonk and Hogan when comparing PPH and IPA.**

We agree on that. Especially, the reference to GCMs were confusing and we agree that the reference to Shonk and Hogan (2008) is irrelevant and confusing at this point. Therefore, we removed it and revised the paragraph as follows:

*"Several independent studies investigated the influence of the plane-parallel assumption on cloud retrievals (e.g. Cahalan, 1994; Loeb and Davies, 1996; Marshak et al., 1998; Zinner et al., 2006; Varnai and Marshak, 2007). They found that the magnitudes of model biases are related to the degree of horizontal photon transport. In 1D radiative transfer simulations clouds are divided into separate vertical columns with horizontal homogeneous optical and microphysical properties (independent pixel approximation, IPA). However, horizontal photon transport cannot be neglected in case of inhomogeneous clouds. Additionally, multiple scattering due to 3D microphysical cloud structures smooth the horizontal radiation field. On small scales, this limits the accuracy of IPA. For example, Cahalan (1994) and Marshak et al. (1995) revealed discrepancies for individual pixel radiances exceeding 50 % due to a plan-parallel bias."*

*"In many remote-sensing applications clouds are assumed as plane-parallel (Francis et al., 1998; Iwabuchi and Hayasaka, 2002; Garrett et al., 2003), which may introduce biases into the modeled radiation budget (Shonk et al., 2011). For example, in the cases of cirrus, Carlin et al. (2002) found a plane-parallel cirrus albedo bias of up to 25 % due to spatial cirrus inhomogeneity. For Arctic stratus over variable sea-ice surfaces, Rozwadowska and Cahalan (2002) reported a plane-parallel albedo bias of less than 2 %, but an absolute value of the transmittance bias that can exceed 10 %."*

2. **Lines 61-68. Here, cloud overlap is mixed into 3D effects and Monte Carlo discussion. Misleading. You can account for overlap perfectly, but still ignore 3D effects by performing IPA calculations on the perfectly overlapped cloud field.**

The reviewer is right. Talking about cloud overlap schemes at this point is misleading. Therefore, we removed those parts, which are related to cloud overlap schemes, in the resubmitted version of the manuscript.

3. **Lines 75-77. I'm sure that ECMWF models do not need the two point statistics of cloud structure (as derived by autocorrelation and power spectrum analysis), but some information on the PDF, i.e., the inhomogeneity parameters of section 3 derived from one point statistics, so invoking "spatial features" "below the meter scale" is again inappropriate.**

In conjunction with the comment above and comments by the other reviewers we removed the reference to ECMWF models in the resubmitted version of the manuscript. We further revised the statement "spatial features below the meter scale" to clarify that this is related to clouds in reality.

*"General circulation or numerical weather forecast models require sub-grid scale parameterizations of, e.g., cloud structures, liquid water content (LWC), and/or ice water content (IWC) (Huang and Liu, 2014). In reality, cloud structures reveal spatial features down to distances below the meter scale (Pinsky and Khain, 2003). Therefore, measurements with appropriate spatial and temporal resolution have to be conducted in order to derive the needed parameterizations. […]"*

4. **Lines 141-142: "However, the fact that rho_tau can exceed values of unity and depends on the average value might lead to misinterpretations." Why? I don't see anything wrong with values greater than unity.**

That is true. There is nothing wrong with values greater than unity. This wording belonged to a former version of the manuscript, where we wanted to say that there is no upper limit for $\rho_\tau$ where the clouds can be found to be inhomogeneous to 100 %. We revised this part by the following:

*"However, $\rho_\tau$ has no predefined upper limit, which might lead to misinterpretations in a variability analysis. This renders $\rho_\tau$ not as a quantitative, but qualitative measure only."*

5. **Lines 157-158: It would be simpler to say that chi is the scaling factor with which mean tau needs to be multiplied to approximate the IPA albedo.**

We have used the suggested wording from the reviewer to simplify this sentence.

*"Thus, the logarithmically averaged $\tau$ provides a way to account for cloud inhomogeneity effects in plane-parallel radiative transfer calculations using $\chi_\tau$ as a scaling factor with which $\tau$ needs to be multiplied to approximate the IPA albedo."*

6. **Page 7: One has to be careful when comparing inhomogeneity parameters across publications. First, pixel size matters. Second, and most importantly, the domain size matters. The bigger your reference domain, the wider the PDF, the larger the inhomogeneity. So this is by no means a trivial comparison.**

We thank the reviewer for this advice. Indeed, a comparison of the results is complicated for different pixel and domain sizes. However, we still like to show and refer the results from other studies. Therefore, we kept this comparison, but we included a paragraph, which points out the restriction of a comparison. The reader has to keep in mind that the results from the different studies are related to different pixel and domain sizes. Furthermore, we included the pixel and domain size of the investigated cases in Table 1 and added the pixel and domain sizes for the literature cases at the relevant parts.

*"The three 1D inhomogeneity parameters $\rho_\tau$, $S_\tau$, and $\chi_\tau$ are calculated for each retrieved field of $\tau_{ci}$ and $\tau_{st}$ from the CARRIBA and VERDI campaigns. The results are listed in the right three columns of Tab. 1. When comparing them to literature values one has to keep in mind that cloud inhomogeneities appear on different spatial scales. E.g., cloud fields may change on synoptic scales (~ 100 km) or dynamic scales (10 – 100 m) depending on the cloud type. Therefore, inhomogeneity parameters depend on the pixel and domain size of the analyzed cloud fields. The larger the domain size or the smaller the pixel size is, the broader the probably density function of the cloud parameter may become. Therefore, a comparison of different cloud cases is only valid when pixel size and cloud domain are in the same range."*

We also revised this statement in the conclusion part of the manuscript:

*"Considering the pixel and domain size of the analyzed measurements, the results from the calculated 1D inhomogeneity parameters $\rho_\tau$, $S_\tau$, are in agreement with values given in the literature for similar cloud types."*

7. **Lines 188-190. It's not a matter of directional dependence only. It's mostly a matter of spatial coherence of cloud condensate, in other words how the variability is distributed across scales.**

That is true. Therefore, we have included this statement to the relevant sentence.

*"Therefore, not only the horizontal inhomogeneity, but also the spatial coherence of cloud inhomogeneity parameters and their directional dependence need to be investigated (Hill et al., 2012)."*

8. **Lines 204-205: Why are negative autocorrelations ambiguous?**

The reviewer is right. The use of the word "ambiguous" is not suitable for the statement we wanted to give here. We revised the paragraph by the following lines to clarify what we wanted to say at this point:

*"Here, only the degree of correlation matters; the positive or negative sign of the autocorrelation result is of less importance. To avoid misinterpretations with the sign, the squared autocorrelation function $P^2_\tau(L_x, L_y)$ is used here.*

**9. Eq. (5): If the scale length is typically defined as the distance at which the autocorrelation drops to 1/e, shouldn't the scale length of squared autocorrelation be defined as the distance where it drops to 1/eˆ2?**

We thank the reviewer for this advice. It is true that $1/e^2$ should be used as a threshold when the squared autocorrelation function is used. We recalculated the decorrelation lengths, revised the relevant figures, tables, and text parts. However, the overall conclusion we made on behalf of the decorrelation lengths has not changed. Only the magnitude of the derived values for the decorrelation lengths have changed. Therefore, here in this reply, we only like to show the revised Figure 4. The remaining changes to the manuscript with regard to new decorrelation length values are marked in the additionally submitted author's response document.

[Figure]

*Fig. 1: Revised Fig. 4 using the new threshold for calculating the decorrelation length.*

**10. Lines 255-256: It's not that they are not well-suited, it's that by themselves they provide incomplete information, i.e., not the whole story.**

We revised this sentence by the reviewer's suggestion.

*"This reveals that the 1D inhomogeneity parameters $\rho_\tau$, $S_\tau$, and $\chi_\tau$ just provide incomplete information for a comparison of different types of clouds as they are not able to consider the horizontal structure of cloud inhomogeneities."*

**11. Power spectrum scale break analysis: Have the authors given any thought on whether the comparisons of scales in terms of physical units (m) makes sense when the pixel sizes are different? With pixel size varying, the extent to which the smoothing is resolved is also different, so I was wondering whether defining the scale lengths in terms of pixel number would bring the results closer together.**

Yes, we had a thought on this. It is right, a comparison of scales in terms of physical units (m) may be difficult, when the pixel sizes are different. However, here in this case they differ not much. Indeed, along the swath (pixel width) the pixel size for the subtropical cirrus cases is twice as large as for the Arctic stratus cases. This is due to the geometry and different

distances between sensor and cloud. However, the pixel size across the swath direction (pixel length), which depends on the integration time, cloud/ aircraft velocity, is almost the same for the measurements during CARRIBA subtropical cirrus) and VERDI (Arctic stratus). The pixel length is about 5±2 m for both cases. Furthermore, the differences are far below the detected scale breaks. Therefore, a comparison is possible in this case.  To better clarify this issue, we added the following lines to the manuscript:

*"[…] Furthermore, both cloud cases, subtropical cirrus and Arctic stratus, exhibit a similar pixel length along $L_y$ (5±2 m), which results from the chosen frame rate (subtropical cirrus: 4 Hz, Arctic stratus: 30 Hz) and given cloud (20 m s$^{-1}$) and aircraft velocity (70 m s$^{-1}$). This allows a direct comparison between these two different cloud types with different observation geometry."*

**Typos and other minor stuff:**

1.  **Line 10: "VERtical" instead of "VERical". Also Line 97.**

Changed to *"VERTical"*

2.  **Lines 120 and 272: I think you wanted to use "fingerprint' rather than "footprint". The term "footprint" in remote sensing indicates the resolution, i.e., pixel size.**

Changed to *"fingerprint"*

3.  **Line 328: "inhomogeneity".**

Changed to *„inhomogeneity"*

4.  **Line 337: "too small".**

Changed to *"too small"*

---

## Author Comment (AC2) · 15 Dec 2016

We thank the reviewer for helpful comments which improved the manuscript significantly. Especially, by adding more explanations the revised manuscript will be easier to understand for the reader. The detailed replies on the reviewers comments are given below and structured as follows. Reviewer comments have bold letters, are labeled, and listed always in the beginning of each answer followed by the author's comments including (if necessary) revised parts of the paper. The revised parts of the paper are written in quotation marks and italic letters.

1. **However, in addition to the comments and suggestions listed below, my main concern with the paper is the lack of a clear statement on what new we were supposed to learn at each step of both (the one-point and the two-points) analyses provided in the paper. What is the main message the authors want us to take home after reading it? I got a feeling that the paper is much more descriptive than conclusive. I'd like to see a list of bullets/statements, at least, in the 'Summary and Conclusion' section.**

The reviewer is right. So far we had been too focused on the feasibility of the study and missed to point out the conclusions of our analysis clearly. We did not use a list of bullets, but we revised the conclusion part to make the most important results more clear.

*"[…] Furthermore, the results from the 2D analysis showed that for the observed cloud cases the subtropical cirrus was more homogeneous than the Arctic stratus. This result was not available from the investigation of the commonly used 1D inhomogeneity parameters. Therefore, using 2D methods in future studies for the characterization of cloud inhomogeneities is advisable, since their information content exceeds the information content of the commonly used 1D inhomogeneity parameters. Nowadays, 2D images of cloud fields are widespread by e.g., measurements of all-sky cameras or satellite observation with high spatial resolution. Applying the presented methods to such continuous measurements would provide detailed views into the climatology of cloud inhomogeneities. […]"*

*"[…] We found such differences for more than the half of the observed cloud scenes. Therefore, the directional structure of cloud inhomogeneities should be taken into account, when cloud inhomogeneities are characterized. Clearly some clouds will have less directional dependence of cloud inhomogeneity, but this does not mean it is not useful to measure it. It is expected that the information content derived from the directional analysis of cloud inhomogeneities can clearly improve sub-grid scale parametrizations in weather and climate models. For this, depending on the application, the decorrelation length (size and structure of cloud inhomogeneities) or the scale breaks (horizontal photon transport, 3D radiative effects) may provide better proxies compared to commonly used 1D inhomogeneity parameters.*

*However, so far only two cloud types were investigated. To build up a better idea on cloud inhomogeneity of different cloud types, more high definition observations of cloud fields are needed. Beside dedicated field campaigns, continuous observations by all-sky cameras or satellites with high spatial resolution such as LandSat (15-90 m resolution) or ASTER (Advanced Spaceborne Thermal Emission and Reflection Radiometer, 15-90 m resolution) may provide the required data.*

*The 1D and 2D autocorrelation functions and Fourier analysis in conjunction with the derived decorrelation length and scale breaks are a helpful tool to verify cloud resolving models in terms of typical horizontal cloud geometries."*

**2. A) The list of references is very rich but, as always, is incomplete. I would definitely add two more very relevant papers. The first one is**

*Davis, A., Marshak, A., Gerber, H., and Wiscombe, W., 1999: Horizontal structure of marine boundary-layer clouds from cm– to km–scales. J. Geophys. Res. 104, 6123-6144.*

**In this paper the authors discuss the structure of marine stratocumulus clouds down to 4-cm scale using both spectral and structure function analyses.**

We added the reference at the point where we are talking about the small-scale break. Please check comment number 3 for our changes .

**B) Another paper is**

*Barker, H.W., B. A.Wielicki, and L. Parker, 1996: A parameterization for computing grid-averaged solar fluxes for inhomogeneous marine boundary layer clouds. Part II: Validation using satellite data. J. Atmos. Sci., 53, 2304–2316.*

**(May be also the Part 1). This paper, I believe, was the first to use the ratio ν=(<τ>/στ)2 to quantify cloud inhomogeneity.**

We have also included a reference to this paper in the current manuscript

*"Therefore, similarly to the studies by Barker et al. (1996), who used ratios between mean $\tau$ and the variance of $\tau$, Davis et al. (1999) and Szczap et al. (2000) utilized the normalized inhomogeneity measure $\rho_\tau$ to quantify the horizontal inhomogeneity of $\tau$."*

**3. Small-scale break ξτ,S. I wonder if the small-scale noise can be reduced by averaging over all cases or over all columns (or rows) in one case. Also, please, compare the location of your small-scale breaks with the once reported by Davis et al. (1999). It was not clear for me what could be learned about cloud structure from the reported small-scale breaks. How does your conclusion depend on pixel size and uncertainty in observations? Please summarize.**

If we average over all columns as proposed by the reviewer the result looks like in Fig. 6 from the original submitted manuscript. The overall appearance of the lines is more smooth then, but their distribution on the y-direction is relatively broad. Therefore, we decided to use the power spectral densities along and across the prevailing direction only for the discussion of the small-scale break. We have also compared the results to the values given by Davis et al. (1999), from which it becomes clear that the small-scale break may have a physical explanation (for scales larger than the pixel size) and is not only related to the white noise, which of course is the reason for the flat power spectral signal for scales below the pixel size. However, since there is a directional behavior we like to keep this paragraph, although it is not possible to fully explain the reason for the small-scale breaks, which are in larger size ranges than the pixel size.

*"[…] Furthermore, the ranges of the derived small-scale breaks $\xi_{\tau,s}$ are found to be close to the ranges of the small-scale breaks reported in literature. Davis et al. (1999) derived small-scale breaks for a broken-stratocumulus/towering cumulus cloud complex from LWC measurements with a particulate volume monitor probe (4 cm resolution) at ranges of about 2-5 m. They proposed that those small-scale breaks are related to extreme values in the detected LWC, which appear on small horizontal scales. Besides Poissonian fluctuations of the cloud optical thickness $\tau$ and the white noise related to power spectral signals at scales below the pixel size this might be a further explanation for the derived small-scale breaks in the current study and needs to be investigated in further studies."*

4.  **A) Large-scale break ξτ,L. After Fig. 7, I'd recommend to mention that CARRIBA ξτ,L << VERDI ξτ,L especially, for the most homogeneous cases of C-02 and C-03. This is partly because ξτ,L, as the radiative smoothing scale, is the harmonic mean of the cloud geometrical thickness and the transport mean free path. Both factors are much smaller for Arctic stratus than for cirrus.**

We thank the reviewer for this advice. We used the reviewer's suggestion and included this information at the end of the relevant paragraph.

*"[…] Furthermore, the resulting large scale breaks $\xi_{\tau,L}$ confirm the results from the derived decorrelation lengths $\xi_\tau$ that the subtropical cirrus observed during CARRIBA is more homogeneous (larger $\xi_\tau$ and $\xi_{\tau,L}$) than the Arctic stratus from VERDI (smaller $\xi_\tau$ and $\xi_{\tau,L}$). This is related to the fact that $\xi_{\tau,L}$, which is the radiative smoothing scale, is a function of the cloud geometrical thickness and the transport mean free path. For Arctic stratus both parameters are significantly smaller than for subtropical cirrus."*

**B) I'd also recommend comparing the theoretical values of the radiative smoothing scale with the observed ones, ξτ,L.**

We have now included a comparison to theoretical values reported by Marshak et al. (1995):

*"[…] Especially the values for the Arctic stratus are in the size range, which was also reported by Marshak et al. (1995), who found scale breaks for fractal clouds in the range of 200-500 m. […]"*

5.  **A) Retrieval of τ. I know that several references on the retrieval processes are given. However, the way τ-field has been retrieved is important for understanding the analysis provided in the paper. The main question is how much the retrieved τ-field is influenced by 3D radiative effects. I'd recommend to briefly describing here the retrieval processes.**

We have now included the most necessary information on the retrieval technique we applied in this study:

*"Simulations are performed with the radiative transfer solver DISORT 2 (Discrete Ordinate Radiative Transfer). Input parameters such as cloud optical properties, aerosol content and spectral surface albedo are provided by the library for radiative transfer calculations (libRadtran, Mayer et al., 2005). The required profiles of thermodynamic parameters are derived from measurements from radiosondes and/or dropsondes. Despite of assuming plane-parallel clouds in the simulations, the investigation of 3D radiative effects is still possible using the retrieved fields of $\tau$, but directional features related to the scattering phase function are avoided. $I_\lambda^\downarrow$ and $I_\lambda^\uparrow$ were simulated as a function of values of $\tau_{ci}$ and $\tau_{st}$, respectively. The simulations were performed for all scattering angles within the FOV of AisaEAGLE. Thus, simulated grids of possible $I_\lambda^\downarrow$ and $I_\lambda^\uparrow$ and corresponding $\tau_{ci}$ and $\tau_{st}$ are available for each time step of the measurements and each spatial pixel. The retrieved $\tau_{ci}$ and $\tau_{st}$ are derived by interpolating the simulated $I_\lambda^\downarrow$ and $I_\lambda^\uparrow$ to the measured value for each spatial pixel using a linear interpolation. More detailed descriptions and sensitivity tests of the applied retrieval procedures are reported by Schäfer et al. (2013) for subtropical cirrus and by Bierwirth et al. (2013) as well as Schäfer et al. (2015) for Arctic stratus. […]"*

**B) Another point, I was not convinced that from analyzing the structure of the retrieved cloud optical depth fields for inhomogeneity, one can learn something new compared to the analysis of the measured fields of radiance. I'd recommend, in parallel to, say, Figs. 5 or 6 (or even 7), showing some results of the analysis of energy spectra for the radiance fields.**

There is one major reason for using cloud optical thickness fields instead of radiance fields in this study. If radiance fields are used instead of cloud optical thickness, the features of the scattering phase function would contaminate the analysis of cloud inhomogeneity structures performed with 2D methods. For one-dimensional analysis using autocorrelation functions or Fourier transformations cloud parcels are observed within the same narrow scattering angle range. Changes in the observed radiance field are then most probably related to the cloud and not to the features of the scattering phase function. However, for 2D images this is different since they cover a wide range of scattering angles. Please find below in Fig. 2 a measurement example extracted from Schäfer et al. (2013), which shows a measured radiance field (left) and the corresponding cloud optical thickness field (right). The radiance plot clearly shows features of the scattering phase function, namely the increase of radiance closer to the Sun (located on the right side of the image). This signature of the radiance field would be imprinted into the autocorrelation or Fourier analysis. In other measurement cases e. g. also halo events or cloud bows could be imprinted in the radiance field. Therefore, we have used the corresponding fields of cloud optical thickness, where those features of the scattering phase function are not included as can be seen in the image on the right side.

[Figure]

Fig. 2: Radiance field (left) and corresponding cloud optical thickness field (right) (Schäfer et al., 2013)

We have also included more information and extended the relevant paragraph to make it more understandable to the reader.

*"As proposed by Marshak et al. (1995), Oreopoulos et al. (2000), or Schröder (2004), horizontal cloud inhomogeneities are studied by scale analysis of cloud-top reflectances. However, radiance measurements include the information of the scattering phase function (e.g., forward/backward scattering peak, halo features) in the measured fields of radiance (Schäfer et al., 2013). To avoid artefacts in the scale analysis resulting from such features, parameters that are independent on the directional scattering of the cloud particles have to be analyzed. The cloud optical thickness $\tau$ does not include the fingerprint of the scattering phase function. Therefore, the ground-based and airborne measured fields of $I^{\downarrow}_{\lambda}$ (CARRIBA) and $I^{\uparrow}_{\lambda}$ (VERDI) were used to retrieve horizontal fields of $\tau$ with a spatial resolution of less than 10 m. The retrieved fields of $\tau$ were then applied to investigate horizontal cloud inhomogeneities of subtropical cirrus (index ci) and Arctic stratus (index st)."*

*"[...] Despite of assuming plane-parallel clouds in the simulations, the investigation of 3D radiative effects is still possible using the retrieved fields of $\tau$, but directional features related to the scattering phase function are avoided. [...]"*

6. **Decorrelation length ξτ**. I wonder why did you use 1/e for the squared autocorrelation function rather than $1/e^2$.

We thank the reviewer for this advice. Now, we have used this new threshold to recalculate the decorrelation lengths, revised the relevant figures, tables, and text parts. However, the overall conclusion we made on behalf of the decorrelation lengths has not changed. Only the magnitude of the derived values for the decorrelation lengths have changed. Therefore, here in this reply, we only like to show the revised Figure 4. The remaining changes to the manuscript with regard to new decorrelation length values are marked in the additionally submitted author's response document.

[Figure]

*Fig. 1: Revised Fig. 4 using the new threshold for calculating the decorrelation length.*

---

## Author Comment (AC3) · 15 Dec 2016

We thank the reviewer for the helpful comments which improved the manuscript significantly. The detailed replies on the reviewer's comments are given below and structured as follows. Reviewer comments have bold letters, are labeled, and listed always in the beginning of each answer followed by the author's comments including (if necessary) revised parts of the paper. The revised parts of the paper are written in quotation marks and italic letters.

**Comments:**

1. **Abstract – there are too many technical terms in the abstract. While reading it, it was not clear to me what the "decorrelation lengths" and "scale breaks" refer to (lines 11 to 14), and without further context, I did not know what the authors meant by "directional cloud inhomogeneities". All became clear having read the paper, although this defeats the point of the abstract – it should be a standalone block of text that is entirely contained. These terms need clarification in the abstract.**

We have revised the Abstract accordingly to make it easier to understand for the reader. Now, we have tried to avoid the technical terms.

*"Clouds exhibit distinct horizontal inhomogeneities of their optical and microphysical properties, which complicate their realistic representation in weather and climate models. In order to investigate the horizontal structure of cloud inhomogeneities, two-dimensional (2D) horizontal fields of optical thickness ($\tau$) of subtropical cirrus and Arctic stratus are investigated with a spatial resolution of less than 10 m. The 2D $\tau$-fields are derived from (a) downward (transmitted) solar spectral radiance measurements from the ground beneath four subtropical cirrus clouds, and (b) upward (reflected) radiances measured from aircraft above ten Arctic stratus clouds. The data were collected during two field campaigns: (a) Clouds, Aerosol, Radiation, and tuRbulence in the trade wInd regime over BArbados (CARRIBA), and (b) VERtical Distribution of Ice in Arctic clouds (VERDI). One-dimensional (1D) and 2D autocorrelation functions, as well as power spectral densities are derived from the retrieved $\tau$-fields. The typical spatial scale of cloud inhomogeneities are quantified for each cloud case. Similarly, the scales at which three-dimensional (3D) radiative effects influence the radiance field are identified. In most of the investigated cloud cases considerable cloud inhomogeneities with a prevailing directional structure are found. In these cases, the cloud inhomogeneities favour a specific horizontal direction while across this direction the cloud is of homogeneous character. The investigations reveal that it is not sufficient to quantify horizontal cloud inhomogeneities by 1D inhomogeneity parameters; 2D parameters are necessarily required."*

2. **Section 1 – the authors cover a great deal of literature in this section, although in places in a somewhat haphazard manner – they may want to have another think about the structure of the section. See following comments.**

According to the reviewer's comments, we have restructured Section 1. Please find our changes below the particular comments.

3. **A) Lines 41 to 42 – to me it is unclear what the sentence "a variability of cirrus albedo of up to 25% due to spatial cirrus inhomogeneity" means. I assume it refers to a plane-parallel bias, although the sentence appears in advance of the discussion of the plane-parallel approximation (from line 46).**

That is true. The necessary information on the plane-parallel bias were missing in this sentence. We revised it as follows:

*"For example, in the cases of cirrus, Carlin et al. (2002) found a plane-parallel cirrus albedo bias of up to 25 % due to spatial cirrus inhomogeneity."*

**B) Lines 46 to 60 – this paragraph is very muddly and confusing. It starts off introducing the plane-parallel approximation, before jumping to the independent pixel approximation, which they say is used in 1D radiative transfer calculations, but it follows after a mention of GCMs, perhaps suggesting that the IPA is used in GCMs. This paragraph should be reworded such that the distinction is clearer. The paragraph continues to cite 50% and 8% errors from two different studies, although no comment is made as to why these are so different.**

We agree on that. Especially, the reference to GCMs were confusing. Furthermore, the reference to Shonk and Hogan (2008) may have been confusing at this point. With respect to other reviewer comments we agreed that this citation was rather irrelevant at this point. Therefore, we removed it and revised the relevant parts.

*"Several independent studies investigated the influence of the plane-parallel assumption on cloud retrievals (e.g. Cahalan, 1994; Loeb and Davies, 1996; Marshak et al., 1998; Zinner et al., 2006; Varnai and Marshak, 2007). They found that the magnitudes of model biases are related to the degree of horizontal photon transport. In 1D radiative transfer simulations clouds are divided into separate vertical columns with horizontal homogeneous optical and microphysical properties (independent pixel approximation, IPA). However, horizontal photon transport cannot be neglected in case of inhomogeneous clouds. Additionally, multiple scattering due to 3D microphysical cloud structures smooth the horizontal radiation field. On small scales, this limits the accuracy of IPA. For example, Cahalan (1994) and Marshak et al. (1995) revealed discrepancies for individual pixel radiances exceeding 50 % due to a plan-parallel bias."*

*"In many remote-sensing applications clouds are assumed as plane-parallel (Francis et al., 1998; Iwabuchi and Hayasaka, 2002; Garrett et al., 2003), which may introduce biases into the modeled radiation budget (Shonk et al., 2011). For example, in the cases of cirrus, Carlin et al. (2002) found a plane-parallel cirrus albedo bias of up to 25 % due to spatial cirrus inhomogeneity. For Arctic stratus over variable sea-ice surfaces, Rozwadowska and Cahalan (2002) reported a plane-parallel albedo bias of less than 2 %, but an absolute value of the transmittance bias that can exceed 10 %."*

**4. Lines 64 to 68 – there is some confusion in this paragraph. It starts with a mention of Monte Carlo simulations, and then suddenly into cloud overlap schemes, which are a completely different area to horizontal cloud inhomogeneity. The authors also assert that Tripleclouds is a cloud overlap scheme – this is not true: it is a method for representing horizontal cloud inhomogeneity in GCMs. Additionally, the scheme is called "Tripleclouds", not "Triplecloud". The following description of Huang and Liu's (2014) method is also not clear. This part of the literature review may need restructuring – as the paper is dealing with metrics of horizontal inhomogeneity, it would be better to introduce methods of accounting for horizontal inhomogeneity in a more clear manner – plus the authors may want to include the Monte-Carlo ICA method of Pincus et al (2003).**

The reviewer is right. Talking about cloud overlap schemes at this point is misleading. Therefore, we removed those parts, which are related to cloud overlap schemes, in the resubmitted version of the manuscript. Furthermore, we followed the suggestion of the reviewer and included the Monte Carlo ICA method of Pincus et al. (2003).

*"3D Monte Carlo radiative transfer simulations account for horizontal photon transport (Barlakas et al., 2016). However, they are costly in terms of computation time and memory (Huang et al., 2014). This renders Monte Carlo radiative transfer simulations inappropriate for the application in operational or global models. Other approaches introduce Monte Carlo integration of independent column approximation (McICA), as proposed by Pincus et al. (2003). McICA is a computational efficient technique for computing domain-averaged broadband radiative flux densities in vertically and horizontally variable cloud fields (Pincus et al., 2003). Improvements compared to the plane-parallel assumption are achieved with this approach, but results are still not as accurate as those from 3D Monte Carlo models. To reduce uncertainties associated with the 1D plane-parallel assumption, Huang et al. (2014) apply spatial autocorrelation functions of cloud extinction coefficients to capture the net effects of sub-grid cloud interactions with radiation. With several orders less computation time, this approach reproduces 3D Monte Carlo radiative transfer simulations with an accuracy within 1 %. However, Huang et al. (2014) assumed perfect knowledge about the spatial correlation functions of cloud extinction coefficients, which underlines the need for measurements of comparable resolved inhomogeneity measures."*

**5. Line 75 – why has the ECMWF mode been singled out as an example here? Surely all models need sub-grid scale parameterisations of cloud structure.**

There is no specific reason why ECMWF has been singled out as an example here. Therefore, in the resubmitted version of the manuscript we have decided to single out no special model.

*"General circulation or numerical weather forecast models require sub-grid scale parameterizations of, e.g., cloud structures, liquid water content (LWC), and/or ice water content (IWC) (Huang and Liu, 2014)."*

**6. Line 84 – the authors say that 1D measures of inhomogeneity can lead to unrealistic results. To describe them as unrealistic does not really feel fair. A 2D metric of horizontal cloud structure is more descriptive of the cloud field, but 1D metrics can still provide information about the "bulk" structure of the cloud field over all angles.**

The reviewer is right. "Unrealistic" is a rather hard wording in this context. Therefore, we have revised this paragraph in the following way:

*"[…] In such a case, 1D observations with LIDAR (light detecting and ranging) or point spectrometers can lead to an underestimation or overestimation of the degree of cloud inhomogeneity of the whole cloud scene. For example, a cloud with a rather inhomogeneous character may be classified as horizontally homogeneous (underestimation of inhomogeneity), if the dominating cloud structure has the same orientation as the cloud observational path. Contrary, the cloud inhomogeneity would be overestimated if the cloud is scanned perpendicular to the major directional structure. […]"*

**7. Lines 90 to 94 – the entire scope and contents of the paper are summed up into a short paragraph of four lines. The contents of the paper could be described in a little more detail. Also, this paragraph (and indeed section 1) is lacking in aims and motivations of the study – indeed, all it says is what the authors will do in their paper, with no science questions. This needs expanding.**

Indeed, the entire scope and contents of the paper were summed up into a quite short paragraph. We revised this part and included more information.

*"In this paper, horizontal $\tau$ fields retrieved from solar spectral radiance measurements are analyzed to quantify horizontal inhomogeneities of two cloud types; subtropical cirrus and Arctic stratus. The information content of 1D and 2D approaches on cloud inhomogeneity analysis are compared to identify their scientific value and limits. In Sect. 3, a statistical evaluation of the horizontal inhomogeneity of the fields of $\tau$ is presented using common 1D inhomogeneity parameters from the literature. Those bulk properties are valid to quantify the overall cloud inhomogeneity, but cannot reproduce spatial inhomogeneities of the cloud field. In Sect. 4, the derived bulk properties from the 1D inhomogeneity parameters are compared to 1D and 2D autocorrelation functions. Finally, in Sect. 5, 1D and 2D Fourier analysis is used to investigate the effect of horizontally cloud inhomogeneities on radiative transfer."*

8. **Lines 99 to 100 – this is not a particularly important point, but it would be great to get a better idea of how the scans are made – for example, are they made from a long time series of 1D scans across a path, or are they full 2D scans that overlap? The papers referred to here probably describe this fully, but it would be nice to have a little more information here.**

Indeed, the measurement technique was described a bit to briefly. Now, we have included more information on that, which should help the reader to better understand how the measurements where performed.

*"[…] Two-dimensional (2D) fields of downward and upward solar spectral radiances ($I^{\downarrow}_{\lambda}$, $I^{\uparrow}_{\lambda}$) were measured from the ground (CARRIBA) and from an aircraft (VERDI). The imaging spectrometer AisaEAGLE (manufactured by Specim Ltd., Finland, Hanus et al., 2008; Schäfer et al., 2013, 2015) was used for the measurements. It is a single-line sensor with a field of view of 37° and 1024 spatial pixels detecting radiation in the wavelength range from 400 nm to 970 nm with a spectral resolution of 1.25 nm full width at half maximum (FWHM). The 2D scans of the cloud scenes are generated from sequential (4 Hz to 30 Hz frame rate) measurements of the single sensor-line, while the target (cloud) moves with the wind (ground-based) or the flying aircraft across this sensor line. Adding up all measured lines behind each other, the 2D scan evolves as an image with a spatial (number of sensor pixels) and temporal (number of recorded frames) axis. Applying the known geometry, integration time, cloud and aircraft velocities, the axes dimensions can be transferred into distances. The 2D images evolved either from the heading of the clouds above the sensor line (ground-based), or by the movement of the sensor-line itself above the clouds (airborne). […]"*

9. **Lines 115 to 121 – the wording of this argument is not particularly clear. I think the authors are saying that optical depth is better to use for this analysis because it is not dependent on scattering angle, while radiance is more directional. This could be better explained.**

Now, we have revised this paragraph to make it better understandable to the reader.

*"As proposed by Marshak et al. (1995), Oreopoulos et al. (2000), or Schröder (2004), horizontal cloud inhomogeneities are studied by scale analysis of cloud-top reflectances. However, radiance measurements include the information of the scattering phase function (e.g., forward/backward scattering peak, halo features) in the measured fields of radiance (Schäfer et al., 2013). To avoid artefacts in the scale analysis resulting from such features, parameters that are independent on the directional scattering of the cloud particles have to be analyzed. The cloud optical thickness $\tau$ does not include the fingerprint of the scattering phase function. Therefore, the ground-based and airborne measured fields of $I^{\downarrow}_{\lambda}$ (CARRIBA) and $I^{\uparrow}_{\lambda}$ (VERDI) were used to retrieve horizontal fields of $\tau$ with a spatial resolution of less than 10 m. The retrieved fields of $\tau$ were then applied to investigate horizontal cloud inhomogeneities of subtropical cirrus (index ci) and Arctic stratus (index st)."*

**10. Line 142 – the authors say that the normalised inhomogeneity parameter can exceed one in some cases, and that this could lead to misinterpretation. I am not sure what they mean by misinterpretation – mathematically, it can happen.**

That is true. There is nothing wrong with values greater than unity. This wording belonged to a former version of the manuscript, where we wanted to say that there is no upper limit for $\rho_\tau$ where the clouds can be found to be inhomogeneous to 100 %. We revised this part by the following:

*"However, $\rho_\tau$ has no predefined upper limit, which might lead to misinterpretations in a variability analysis. This renders $\rho_\tau$ not as a quantitative, but qualitative measure only."*

**11. Page 7 – through much of this page, the authors compare the different measures of 1D inhomogeneity from their study with values from other studies and say that they "compare well" and similar. It should be noted that these inhomogeneity parameters can be highly related to the domain and pixel size of the observations in the studies (see Shonk et al 2010, part one). To say that values "compare well" is probably only fair to use when comparing with studies that use very similar domain/pixel sizes. The same is also true in the conclusions in lines 398 to 399.**

We thank the reviewer for this advice. Indeed, a comparison of the results is complicated for different pixel and domain sizes. However, we still like to show and refer the results from other studies. Therefore, we kept this comparison, but we included a paragraph, which points out the restriction of a comparison. The reader has to keep in mind that the results from the different studies are related to different pixel and domain sizes. Furthermore, we included the pixel and domain size of the investigated cases in Table 1 and added the pixel and domain sizes for the literature cases at the relevant parts.

*"The three 1D inhomogeneity parameters $\rho_\tau$, $S_\tau$, and $\chi_\tau$ are calculated for each retrieved field of $\tau_{ci}$ and $\tau_{st}$ from the CARRIBA and VERDI campaigns. The results are listed in the right three columns of Tab. 1. When comparing them to literature values one has to keep in mind that cloud inhomogeneities appear on different spatial scales. E.g., cloud fields may change on synoptic scales (~ 100 km) or dynamic scales (10 – 100 m) depending on the cloud type. Therefore, inhomogeneity parameters depend on the pixel and domain size of the analyzed cloud fields. The larger the domain size or the smaller the pixel size is, the broader the probably density function of the cloud parameter may become. Therefore, a comparison of different cloud cases is only valid when pixel size and cloud domain are in the same range."*

We also revised this statement in the conclusion part of the manuscript:

*"Considering the pixel and domain size of the analyzed measurements, the results from the calculated 1D inhomogeneity parameters $\rho_\tau$, $S_\tau$, are in agreement with values given in the literature for similar cloud types."*

12. **Line 204 – the authors take the square of the autocorrelations to get an absolute correlation, which is fine, although I would contend that finding negative correlations is not necessarily ambiguous.**

The reviewer is right. The use of the word "ambiguous" is not suitable for the statement we wanted to give here. We revised the paragraph by the following lines to clarify what we wanted to say at this point:

*"Here, only the degree of correlation matters; the positive or negative sign of the autocorrelation result is of less importance. To avoid misinterpretations with the sign, the squared autocorrelation function $P^2_\tau(L_x,L_y)$ is used here. "*

13. **Line 209 – if the positive and negative signs indicate the shifting direction, I would say they do have physical meaning.**

To avoid confusion, we revised the part by the following sentence:

*"The positive and negative signs of $L_x$ and $L_y$ in Fig. 2b and 2d illustrate the direction into which one the particular image is shifted against itself to derive the depicted autocorrelation coefficients."*

14. **Figure 3 – this caption could do with a little more information. The caption is entirely written in mathematical terms – it is clear what these mean by reading the text, but defining them briefly in the caption could be helpful.**

We have extended the content of the figure caption with more information and also included the words for the single symbols to make it better readable.

*"Figure 3. (a) Average of squared 1D autocorrelation functions $P^2_\tau(L_x;L_y)$ (solid lines), calculated for pixels, which are orientated into the direction of the blue ($L_x$) and red line ($L_y$) illustrated in Fig. 2b. The dashed lines mark the derived distance of the decorrelation length $\xi_\tau$, where $P^2_\tau(L_x)$ and $P^2_\tau(L_y)$ are decreased to $1/e^2$. (b) Same as (a) for $P^2_\tau(L_x;L_y)$ shown in Fig. 2d."*

15. **Figure 4 – why are the red and blue bars stacked? This gives the impression that the authors are summing together the Lx-direction and Ly-direction decorrelation length scales – by the looks of Table 2, this is not the case. I assume they have been arranged in this way because the blue bar is always taller, implying that the decorrelation length in the Ly-direction is always greater. (Which itself is an interesting question that is not addressed.)**

The reviewer is right. We revised this figure (also with respect to comments of the other reviewers) and put the bars next to each other. Please find the revised Fig. 4 below. The reason for the systematic larger blue bars in each measurement case is that these are the calculations for the decorrelation lengths along the prevailing directional structure of the

cloud inhomogeneities, while the red bars are the calculations for the decorrelation lengths across those prevailing directional structures. To make this more clear we revised a few text parts:

*"The decorrelation lengths are calculated for each measurement case of CARRIBA and VERDI along ($\xi_\tau^{\updownarrow}$) and across ($\xi_\tau^{\leftrightarrow}$) the prevailing directional structure of the cloud inhomogeneities, which is identified by the 2D autocorrelation analysis. […]"*

*"[…] Furthermore, the results indicate that for most of the measurement cases from CARRIBA and VERDI a distinct directional structure of cloud inhomogeneities is observed. The results for $\xi_\tau^{\updownarrow}$ are in 9 of the 14 investigated cases more than twice as large as for $\xi_\tau^{\leftrightarrow}$."*

[Figure]

Revised Fig. 4

**16. Lines 256 to 257 – the authors provide a conclusion for section 4 with the sentence "A comparison can only indicate … inhomogeneities." Many aspects of this sentence are unclear – indeed, I am not sure what the authors are concluding.**

The sentence was probably too short with too less information. Now we have extended this part by the following:

*"However, the absolute values of $\xi_\tau^{\updownarrow}$ and $\xi_\tau^{\leftrightarrow}$ for the Arctic stratus are smaller (more inhomogeneous) than those for the subtropical cirrus, although the 1D inhomogeneity parameters from Tab. 1 yield similar values. This reveals that the 1D inhomogeneity parameters $\rho_\tau$, $S_\tau$, and $\chi_\tau$ just provide incomplete information for a comparison of different types of clouds as they are not able to consider the horizontal structure of cloud inhomogeneities. Differences can only be observed by an evaluation of the horizontal pattern of the cloud inhomogeneities."*

**17. Lines 348 to 352 – the authors introduce the reduced noise terms, but again perhaps a little too technically for someone who does not know how to do it – they may want to consider revising these few lines to make the method clearer.**

We have now included the relevant equations for the octave binning. This should help the reader to understand the method easier.

*"To evaluate the resulting 1D Fourier spectra with reduced noise (rn) characteristics, the $E_{rn}(k) \sim k^{\beta}$ are calculated with the use of octave binning, following the method proposed by Davis et al. (1996), Harris et al. (1997), and Schröder (2004). Logarithmically spaced bins $k_n$ are calculated by:*

$$k_n = \frac{1}{2^n} \sum_{i=2n}^{2^{n+1}-1} k_i, n = 1, 2, ..., \log_2(N-2),$$

*for the number of data points N. $E(k_{rn})$ is then obtained by:*

$$E(k_{rn}) = \frac{1}{2^n} \sum_{i=2n}^{2^{n+1}-1} E(k_i), n = 1, 2, ..., \log_2(N-2).$$

*Within each bin $2^n$ data points are averaged. In addition to the reduced noise of $E(k_{rn})$ compared to $E(k_y)$ the binning provides a uniform contribution of all scales to the average values. The $E(k_{rn})$ derived from the octave binning are included as green diamonds in Fig. 6. The data of the octave binning were used to fit the spectra for different slopes in the different scale ranges."*

**18. Line 375 – is the CARRIBA cirrus "more homogeneous" than the VERDI stratus? Earlier, the authors say that cases C-02 and C-03 are more homogeneous than C-01 and C-04, and in Table 2, we see that they have larger decorrelation length scales. But the decorrelation scales from the VERDI stratus are smaller than those from the CARRIBA cirrus, which surely means that the cirrus are more inhomogeneous than the stratus? Or am I missing something?**

We admit that this might be confusing, but the subtropical cirrus is more homogeneous than the Arctic stratus. Like the reviewer said for the subtropical cirrus cases, larger decorrelation lengths are related to more homogeneous clouds (Ci-02 and Ci-03 have a larger decorrelation length than Ci-01 and Ci-04 and are more homogeneous). The Arctic stratus cases in turn have an overall smaller decorrelation length than the subtropical cirrus cases for the spatial domain investigated here. Thus, they are more inhomogeneous than the subtropical cirrus cases. This is related to the fact that homogeneous clouds show a similar horizontal structure over a larger spatial range than inhomogeneous clouds. Therefore, from the autocorrelation analysis the inhomogeneous clouds are decorrelated at smaller spatial scales than the homogenous clouds, which relates to smaller decorrelation length.

19. **Line 408 – I think this first sentence is saying that, when determining cloud inhomogeneity, it is important to consider the full horizontal structure of clouds via a 2D metric rather than a 1D metric. The authors may want to revisit this sentence – at the moment, it seems to imply that performing 2D analysis shows that 1D analysis works.**

It is true. The sentence was written the wrong way around. We revised those lines. Now the message should become clear.

*"However, the comparison of the 2D analysis of squared autocorrelation functions $P^2_\tau(Lx;Ly)$ with the 1D inhomogeneity parameters $\rho_\tau$, $S\tau$, and $\chi_\tau$ showed that it is important to consider the full horizontal structure of clouds using 2D analysis rather than 1D analysis, when determining cloud inhomogeneity."*

20. **Lines 443 to 444 – the final conclusion seems to be that, despite having gone to all the trouble of calculating several different measures of 1D and 2D inhomogeneity, "the directional structure of cloud inhomogeneities generally should be taken into account". This feels like a weak concluding comment. For a start, what is meant by "generally"? Clearly some clouds will have less directional dependence of cloud inhomogeneity, but this does not mean it is not useful to measure it. How is best to determine this two-dimensional inhomogeneity? Which of the methods compared performed best? Which is easiest to calculate? Which is most useful if, say, 2D inhomogeneity parameters were to be included in a climate model? Also, how easy is it to obtain observational cloud data to extract these 2D cloud fields – are they readily available, or should we be carrying out more of these field campaigns to build up a better idea of cloud inhomogeneity? And is there any cause to investigate 3D cloud inhomogeneity metrics? These are a few questions that popped up in my mind having read the paper – it is disappointing that the authors have not included any such consideration on the implications of these results on the scientific community, and how the work could be taken forward.**

The reviewer is right. So far in the manuscript we had included to less information on the feasibility of the study and possible applications of the presented investigations. We revised the conclusion part to make the most important results more clear and we further tried to address the questions raised by the reviewer.

*"[…] Furthermore, the results from the 2D analysis showed that for the observed cloud cases the subtropical cirrus was more homogeneous than the Arctic stratus. This result was not available from the investigation of the commonly used 1D inhomogeneity parameters. Therefore, using 2D methods in future studies for the characterization of cloud inhomogeneities is advisable, since their information content exceeds the information content of the commonly used 1D inhomogeneity parameters. Nowadays, 2D images of cloud fields are widespread by e.g., measurements of all-sky cameras or satellite observation with high spatial resolution. Applying the presented methods to such continuous measurements would provide detailed views into the climatology of cloud inhomogeneities. […]"*

*"[…] We found such differences for more than the half of the observed cloud scenes. Therefore, the directional structure of cloud inhomogeneities should be taken into account, when cloud inhomogeneities are characterized. Clearly some clouds will have less directional dependence of cloud inhomogeneity, but this does not mean it is not useful to measure it. It is expected that the information content derived from the directional analysis of cloud inhomogeneities can clearly improve sub-grid scale parametrizations in weather and climate models. For this, depending on the application, the decorrelation length (size and structure of cloud inhomogeneities) or the scale breaks (horizontal photon transport, 3D radiative effects) may provide better proxies compared to commonly used 1D inhomogeneity parameters.*

*However, so far only two cloud types were investigated. To build up a better idea on cloud inhomogeneity of different cloud types, more high definition observations of cloud fields are needed. Beside dedicated field campaigns, continuous observations by all-sky cameras or satellites with high spatial resolution such as LandSat (15-90 m resolution) or ASTER (Advanced Spaceborne Thermal Emission and Reflection Radiometer, 15-90 m resolution) may provide the required data.*

*The 1D and 2D autocorrelation functions and Fourier analysis in conjunction with the derived decorrelation length and scale breaks are a helpful tool to verify cloud resolving models in terms of typical horizontal cloud geometries."*

**Specific comments:**

1. **Line 9 and 97 – should "VERical" read "VERtical" (or is it spelt like that in the project name)?**

Changed to *"VERTical"*

2. **Line 105 – ""…full width at…", not "…full with at…".**

Changed to *"full width at…"*

3. **Table 1 – the four cirrus cases are all labelled "C-01" – should these be "C-01", "C-02", and so on?**

We corrected the numbering. We further renamed it to Ci-01,… and St-01,…

4. **Line 192 – "both dimensions", not "both dimension".**

Changed to *"both dimensions"*

**5. Figure 4 – "Decorrelation" on the y-axis has been written as "De-Correlation".**

Changed to *"Decorrelation"*

**6. Line 239 and 247 – the value of 3154 m is shown as 3153 m in Figure 3.**

The calculated value was 3153.8 m. in the text we have used the rounded value 3154 m. In the Figure just the decimal place was cut. However, using the new threshold of $1/e^2$ for the calculation of the decorrelation lengths the, the whole value has changed and was revised.

**7. Line 262 – "in a horizontal direction", not "in horizontal direction".**

Changed to *"in a horizontal direction"*

**8. Lines 304 to 305 – "Ea and Eb" has been written as "Ea and Ea".**

Changed to *"$E_a$ and $E_b$"*

**9. Lines 328 and 329 – "inhomogeneity" has been written twice as "inhomogeniety".**

Both changed to *"inhomogeneity"*

**10. Line 337 – "too small", not "to small".**

Changed to *"too small"*

**11. Line 346 – "overlaid", not "overlayed".**

Changed to *"overlaid"*

**12. Line 360  "decreases to 2.2" should be "decreases to –2.2" (based on the numbers on Figure 6).**

Changed to *"-2.2"*

**13. Figure 7 – the caption seems to be a copy of the caption of Figure 6.**

We have revised this mistake.

*"Figure 7. Across swath derived large scale breaks $\xi_{\tau,L}$ for the retrieved fields of $\tau$ from the (a) CARRIBA (Ci- 01 to Ci-04, red) and (b) VERDI (St-01 to St-10, blue) campaigns. The values were derived by using the method presented in Fig. 6."*

**14. Line 408, 411 and 435 – the authors switch here between P and $P^2$ – is there any reason for this?**

We used $P$ and not $P^2$ at the relevant parts because at this point we talked about general properties of autocorrelation functions. At the point where it comes to the decorrelation length we changed it to $P^2$. However, since this may be confusing we revised the relevant parts and directly talk about $P^2$.

**15. Line 473 – "Green-function", not "green-function" (I think they are named after someone called Green).**

Changed to *"Green-function"*

**16. Line 480 – "absorption", not "absorbtion".**

Changed to *"absorption"*

**17. Lines 524 and 526 – check the spelling of "Schroeder" vs "Schröeder".**

Both changed to *"Schröder"*

**18. Line 538 – "Earth's", not "Earths".**

Changed to *"Earth's"*

**19. Finally, the authors have started several sentences throughout the paper with numbers and variables – this is something that may get picked up later in the review process, but they may want to revise these sentences so that they start with words.**

We have checked the whole manuscript for such cases and reworded the sentences, which started with a number or variable.